

# Limb myology and muscle architecture of the Indian rhinoceros *Rhinoceros unicornis* and the white rhinoceros *Ceratotherium simum* (Mammalia: Rhinocerotidae)

Cyril Etienne[1], Alexandra Houssaye[1] and John R. Hutchinson[2]

[1] UMR 7179 Mécanismes adaptatifs et évolution (MECADEV), Centre National de la Recherche Scientifique, Muséum National d'Histoire Naturelle, Paris, France
[2] Structure and Motion Laboratory, Royal Veterinary College, Hatfield, United Kingdom

## ABSTRACT

Land mammals support and move their body using their musculoskeletal system. Their musculature usually presents varying adaptations with body mass or mode of locomotion. Rhinocerotidae is an interesting clade in this regard, as they are heavy animals potentially reaching three tons but are still capable of adopting a galloping gait. However, their musculature has been poorly studied. Here we report the dissection of both forelimb and hindlimb of one neonate and one adult each for two species of rhinoceroses, the Indian rhinoceros (*Rhinoceros unicornis*) and the white rhinoceros (*Ceratotherium simum*). We show that their muscular organisation is similar to that of their relatives, equids and tapirs, and that few evolutionary convergences with other heavy mammals (e.g. elephants and hippopotamuses) are present. Nevertheless, they show clear adaptations to their large body mass, such as more distal insertions for the protractor and adductor muscles of the limbs, giving them longer lever arms. The quantitative architecture of rhino muscles is again reminiscent of that of horses and tapirs, although contrary to horses, the forelimb is much stronger than the hindlimb, which is likely due to its great role in body mass support. Muscles involved mainly in counteracting gravity (e.g. *serratus ventralis thoracis*, *infraspinatus*, *gastrocnemius*, *flexores digitorum*) are usually highly pennate with short fascicles facilitating strong joint extension. Muscles involved in propulsion (e.g. *gluteal muscles*, *gluteobiceps*, *quadriceps femoris*) seem to represent a compromise between a high maximal isometric force and long fascicles, allowing a reasonably fast and wide working range. Neonates present higher normalized maximal isometric force than the adults for almost every muscle, except sometimes for the extensor and propulsor muscles, which presumably acquire their great force-generating capacity during the growth of the animal. Our study clarifies the way the muscles of animals of cursorial ancestry can adapt to support a greater body mass and calls for further investigations in other clades of large body mass.

Corresponding author
Cyril Etienne,
cyril.etienne@cri-paris.org

## INTRODUCTION

Land mammals must support and move the weight of the entire body with their limbs, driven by the muscle-tendon units (e.g., *Hildebrand, 1982*; *Biewener & Patek, 2018*). In ungulates, the forelimb and hindlimb each have a specific role: the forelimb, through its cranial position, tends to support about 60% of body weight and acts mainly in deceleration during steady-state locomotion, whereas the hindlimb has a smaller supportive role but a major propulsive one (*Herr, Huang & McMahon, 2002*; *Witte, Knill & Wilson, 2004*; *Payne et al., 2005*; *Dutto et al., 2006*; *Ren et al., 2010*; *Biewener & Patek, 2018*).

Ungulates vary greatly in terms of mass and general proportions (e.g. a hippopotamus vs. a giraffe vs. a gazelle, *Wilson & Mittermeier, 2011*). Their limb muscles thus vary in organisation (i.e. qualitative myology, notably where each muscle inserts on the bones), architecture (i.e. quantitative geometry of muscle fascicles, including e.g. fascicle length and pennation angle) and ultimately their general functional roles (*Hildebrand et al., 1985*; *Biewener & Patek, 2018*). For a given force, a muscle with a line of action close to a joint will typically generate a weaker moment due to a decreased moment arm, but the velocity of the movement, as well as its range of motion, will be increased (*McClearn, 1985*; *Gans & Gaunt, 1991*; *Pandy, 1999*). This is useful for cursorial animals which rely on speed, but less useful for heavy animals (i.e. several tons) which counteract their body weight with large moments and forces (*Biewener, 1989*; *Biewener & Patek, 2018*).

Muscle architecture is commonly described using several parameters (*Alexander, 1974*; *Gans & De Vree, 1987*; *Payne et al., 2005*; *Payne, Veenman & Wilson, 2005*; *Myatt et al., 2012*; *Cuff et al., 2016*; *MacLaren & McHorse, 2020*). These include muscle mass and total belly length, the length of tendons and fascicles in the muscle, and the pennation angle of the fascicles relative to the line of action. These parameters can be used, for example, to estimate the muscle's physiological cross-sectional area (PCSA), which in turn can be used to estimate the maximal isometric force output of the muscle (*Powell et al., 1984*; *Lieber & Ward, 2011*). Thus, quantitative muscle architecture of different groups of muscles can tell us much about an animal's potential limb functions. Parallel-fibred muscles have a greater working range than pennate muscles, but the latter have the advantage of being able to generate a greater force for the same muscle volume (*Hildebrand et al., 1985*; *Biewener, 1990*; *Azizi, Brainerd & Roberts, 2008*; *Biewener & Patek, 2018*). The organisation and architecture of the locomotor muscles of a species will represent a compromise between all those characteristics suiting the morphology and behaviour of that species, and taking into account its ancestry. Body mass in particular has a major impact on muscle architecture, because a muscle's maximal force output is a function of its cross-sectional area (scaling with linear dimensions squared), whereas mass increases proportionally to the volume of the animal (scaling with linear dimensions cubed; *Biewener, 1989*, *2005*). In large animals, particular adaptations of the musculoskeletal system such as changes in limb posture, bone shape and muscle organisation and architecture become necessary (*Alexander, 1985*; *Biewener, 1989*, *2005*).

Among large mammals, Rhinocerotidae comprises five extant species ranging from an average of 700 kg for *Dicerorhinus sumatrensis*, the Sumatran rhinoceros; to 2,000 kg for *Rhinoceros unicornis*, the Indian rhino and 2,300 kg for *Ceratotherium simum*, the white rhino (*Silva & Downing, 1995*; *Dinerstein, 2011*). The latter two species include adults exceeding three tons. Due to their heavy weight, rhinos have been described as graviportal, along with elephants and hippos (*Hildebrand, 1982*; *Eisenmann & Guérin, 1984*; *Alexander & Pond, 1992*). However, rhinoceroses present marked functional differences from elephants and hippos. Rhinos are all capable of attaining a full gallop, with a suspended phase where all four limbs are off the ground, reaching up to an estimated ~7+ ms$^{-1}$ for *C. simum* and ~12 ms$^{-1}$ for the lighter *Diceros bicornis*, the black rhinoceros (*Garland, 1983*; *Alexander & Pond, 1992*), although empirical studies are very scarce. *Hippopotamus* and elephants cannot adopt a galloping gait (*Dagg, 1973*). Rhinoceros limbs are not as columnar as those of walking elephants, and still present a noticeable flexion of all joints when standing at rest (*Christiansen & Paul, 2001*). This has led other studies to avoid their characterization as graviportal and classify them as mediportal instead, an intermediate category being defined by limbs primarily adapted for weight-bearing but incorporating some cursorial adaptations as well, commonly retained from a cursorial ancestor (*Coombs, 1978*; *Becker, 2003*; *Becker et al., 2009*).

The unusual form and function of rhinoceros limbs emphasise the need for a comprehensive anatomical study of their limb muscles, to better understand how their limbs sustain their large body weight. This would complement the extensive work recently performed on the morphology of rhinoceros limb bones (*Mallet et al., 2019*, *2020*; *Mallet, 2020*; *Etienne et al., 2020*). In terms of both qualitative myology and quantitative architecture, rhinoceroses have been poorly studied. *Haughton (1867)* studied the limbs of a rhinoceros of two or three years old, captured from the wild near Calcutta and acquired by the Dublin zoo, and reported the mass of the individual muscles. It was likely an Indian rhinoceros (*Rhinoceros unicornis*), although the Javan (*R. sondaicus*) and Sumatran rhinoceroses may still have lived near that region at the time (*Foose, Khan & Van Strien, 1997*; *De Courcy, 2010*). *Beddard & Treves (1889)* qualitatively studied two adult Sumatran rhinoceroses (*Dicerorhinus sumatrensis)*, the lightest of all the living rhinos (*Dinerstein, 2011*). No detailed quantitative study of the limb muscles is available. *Alexander & Pond (1992)* provided a few quantitative anatomical details for biomechanical analysis based on bone measurements and video analyses of a running white rhino (*C. simum*). In terms of myology, rhinos' relatives among the Perissodactyla, i.e. tapirs and equids, are more well-known, although tapirs lack a quantitative characterization of their hindlimb (e.g. *Murie, 1871*; *Campbell, 1936*; *Bressou, 1961*; *Barone, 1999*, *2010*; *Brown et al., 2003*; *Payne et al., 2005*; *Payne, Veenman & Wilson, 2005*; *Crook et al., 2008*; *Borges et al., 2016*; *Pereira et al., 2017*). The musculature of the other heaviest mammals, i.e. elephants and hippopotamuses, has been studied qualitatively, but never quantitatively (*Miall & Greenwood, 1878*; *Eales, 1928*; *Mariappa, 1986*; *Weissengruber & Forstenpointner, 2004*; *Fisher, Scott & Naples, 2007*; *Fisher, Scott & Adrian, 2010*; *Trenkwalder, 2013*; *Nagel et al., 2018*).

Here we provide a description of the organization of the limb muscles of two species of rhinoceroses, and a quantitative characterisation of the architecture of those muscles, based on dissections of *Ceratotherium simum* and *Rhinoceros unicornis*. Those two species present a similar average mass, averaging around two tons (*Silva & Downing, 1995*; *Dinerstein, 2011*); as such a large body mass induces an extremely high adaptive pressure (*Hildebrand et al., 1985*; *Biewener, 1989*, *1990*; *Biewener & Patek, 2018*), we might expect it to drive most of the muscular phenotype of our two species and thus to find few differences between them. However, the two species present several differences, like a different body profile: *C. simum* has a low-hanging head whereas *R. unicornis* carries its head higher (*Dinerstein, 2011*). They also display notable differences in limb bone shape (*Guérin, 1980*; *Mallet et al., 2019*, *2020*; *Etienne et al., 2020*), and they live in different habitats, *C. simum* preferring open flatlands while *R. unicornis* is found in semi-open floodplains, swimming easily (*Dinerstein, 2011*). *C. simum* usually displays size dimorphism, with males larger than females, whereas *R. unicornis* displays dimorphism only in captivity, not in the wild, although size dimorphism in rhinos is difficult to quantify (*Dinerstein, 2011*). Therefore, we might still find some differences between our species that could be linked to their differences in morphology and habitat. At a larger scale, we expect that rhino musculature will share features linked to fast running with their close relatives, tapirs and equids; e.g. fast protractor muscles for all limbs and forceful propulsive muscles in the hindlimb, perhaps inherited from early perissodactyls (*Radinsky, 1966*; *Gould, 2017*). However, we expect rhinos, unlike their cousins, to show adaptations to sustain their large body mass that they might share through convergent evolution with *Hippopotamus* and elephants, mainly stronger extensor muscles, particularly in the forelimb, to counteract gravity. Finally, we expect neonate rhinoceroses' muscles to have a much greater relative force-generating capacity than those of adults, because ontogenetic scaling tends to render smaller animals relatively stronger (*Carrier, 1995*, *1996*; *Herrel & Gibb, 2006*).

## MATERIALS & METHODS

### Material

Four specimens of rhinoceroses were dissected in this study (Table 1): two white rhinos (*C. simum*) and two Indian rhinos (*R. unicornis*). For each species we studied a neonate and a female adult of around 40 years of age at death. All specimens died of natural causes or were euthanised by zoos for health issues unrelated to this study. For the adults, the limbs were separated from the carcass at the time of death and frozen until dissection; the neonates were frozen whole (−20 °C). They were all thawed at 4 °C for at least two days before starting to dissect. The specimens were dissected at the Royal Veterinary College, Hawkshead campus, UK; only the left limbs were dissected except for the neonate *R. unicornis* for which we dissected the right limbs.

### Dissections

The skin and superficial fascia were first removed to expose the surface muscles. Each muscle was identified, labelled, photographed and carefully dissected from origin to
**Table 1 Rhinoceros specimens studied.**

| Species | Age | Body mass | Sex | Condition | Origin |
|---|---|---|---|---|---|
| *Ceratotherium simum* | >40 year | 2,160 kg | F | Weight loss and generalized weakness | ZSL Whipsnade Zoo, UK |
| *Ceratotherium simum* | 0 year | 47 kg | M | Stillborn | Details lost (European zoo) |
| *Rhinoceros unicornis* | 38 year | 2,065 kg | F | Ataxia | Woburn Safari Park, UK |
| *Rhinoceros unicornis* | 0 year | 43 kg | Unknown | Stillborn | Munich Hellabrunn Zoo, Germany |

**Note:**
The adult specimens were weighed at death. Both neonates were weighed after thawing and evisceration.

insertion, including any tendon, which was then separated from the muscle belly. Muscle bellies and tendons were cleaned of fat and aponeuroses, weighed using electronic scales to the nearest 0.1 g, and measured using a measuring tape (±1 mm, adults) or digital callipers (±0.1 mm, neonates) from the proximal to the distal end. Muscle fascicles were exposed by cutting along the length of the belly in multiple locations, and their lengths measured at random intervals within the muscle belly. Between three and 10 measurements were made for each muscle for repeatability, with more measurements for larger muscles. Pennation angles of fascicles were also measured using a protractor (±5°); again, between three and 20 measures were taken depending on the muscle and its size.

## Insertion areas

Origin and insertion areas of all the muscles were estimated mainly by observation of the in situ photographs, and occasionally by comparisons with previous works on rhinos (*Haughton, 1867*; *Beddard & Treves, 1889*) as well as what is known in horses from *Barone (1999, 2010)*. Considering that we studied two species of rhinos, the insertion areas are not meant to be species-specific but rather a consensus of what is observed in adult rhinocerotids. If differences between our two species were noted, they were reported.

## Quantitative parameters

Muscle volume was estimated by dividing its mass by a density of 1.06 g cm$^{-3}$ (*Mendez & Keys, 1960*; see also e.g. *Brown et al., 2003*; *Payne et al., 2005*; *MacLaren & McHorse, 2020*). Average fascicle length (AFL) and pennation angle for each muscle were calculated. PCSA was calculated using the following formula:

$$PCSA = \frac{Muscle\ mass * \cos(pennation\ angle)}{density * AFL}$$

The maximal isometric force (Fmax) capacity of each muscle was estimated by multiplying the PCSA by the maximal isometric stress of vertebrate skeletal muscle (300 kPa (*Woledge, Curtin & Homsher, 1985*)). This value was then normalized by dividing it by the weight of the animal (in Newtons; = body mass × 9.81 m s$^{-2}$). The AFL was also normalized by dividing it by the mean of the AFL of all the muscles in the limb. This allowed comparisons of Fmax and AFL between specimens of different masses, particularly between adults and neonates. Normalized Fmax was compared between the

species and the developmental stages using a Student's t-test with the logarithm of the values, using the stats.ttest_ind function of the SciPy Python package (see File S1 for code). If the value for a muscle was missing in any of the two specimens that were compared with the t-test, the muscle was removed in the other specimen compared as well, in order to compare identical sets of muscles. This was the case for eight muscles out of 63 when comparing between both adults, 20 when comparing between both neonates, 11 when comparing both *C. simum* individuals, and 20 when comparing both *R. unicornis* specimens.

## RESULTS

In the Results section, we start by making comparisons of qualitative myology between rhinos and their close relatives among perissodactyls (i.e. tapirs and equids). Hippopotamuses and elephants are included as well, because they share with rhinoceroses a large body mass and might thus present similar size-related adaptations. When relevant, large bovids are also included in the comparisons. We then report on the quantitative architecture of the limb muscles of our four specimens.

### Comparative anatomy of the limb muscles

#### Forelimb

The anatomy of each muscle of the forelimb was recorded (Table 2, Figs. 1, 2), and their origin and insertion on the bones were determined (Figs. 3, 4, 5). Several muscles were damaged (e.g. during limb removal at post-mortem site) and their quantitative parameters could not be measured. These were the *rhomboidei* (RHB) and the *extensor carpi radialis* (ECR) in the adult *R. unicornis*, and the *serrati ventrales* (SV) in the neonate *R. unicornis*. Some muscles were not found at all in some specimens, these were the *brachialis* (BR) and *flexor carpi ulnaris* (FCU) in the adult *R. unicornis*, the *extensor carpi obliquus* (ECO) in the neonate *R. unicornis*, the *brachioradialis* (BRA) in the neonate *C. simum* and the *tensor fasciae antebrachiae* (TFA) in both neonates. We found that muscles were often less clearly differentiated in neonate rhinos. The *serrati ventrales* could not be separated into the *pars cervicis* (SVC) and the *pars thoracis* (SVT) in both neonates but were distinct in both adults. The same applied to the *pars acromialis* (DLA) and *pars scapularis* (DLS) of the *deltoideus* (DL) in the neonate *C. simum*, and the cranial and caudal parts of the *coracobrachialis* (CB) in both neonates. The four *pectorales* were all present, but were difficult to separate in neonates again, especially the two *pectorales superficiales* (the *pectoralis descendens* and the *pectoralis transversus*, PCD and PCT) and the two *pectorales profundi* (the *pectoralis ascendens* and the *subclavius*, PCA and SU). The *anconeus* (AN) was merged with the *triceps brachii caput mediale* (TM) in all specimens except the neonate *R. unicornis*. The *flexor carpi radialis* (FCR) and *flexor carpi ulnaris* were also impossible to separate in the neonates. The ulnar head of the *flexor digitorum profundus* (FDPF) was well differentiated in adult rhinoceroses, but not in neonates. The *pronator teres* was identified only in the adult *C. simum* as a reduced strip, almost entirely tendinous. *Mm. teres minor, palmaris longus, pronator quadratus, supinator* and *extensor pollicis longus et indicis* were not found in any specimen.

**Table 2 General origins and insertions of the muscles of the forelimb in rhinoceroses, with their main action(s) (anatomically estimated function, based on *Barone, 2010*).**

| Name | Abb. | Origin | Insertion | Action |
|---|---|---|---|---|
| *M. omotransversarius* | OT | Wing of the atlas, and likely transverse processes of the first cervical vertebrae | Unclear, most likely distal part of scapular spine and craniomedial humerus proximal to *brachiocephalicus* | Forelimb protraction |
| *M. brachiocephalicus* | BC | Mastoid process of temporal bone | Proximo-cranial aspect of the humeral crest | Neck flexion and rotation, forelimb protraction |
| *M. pectoralis descendens* | PCD | Manubrium, sternum and costal cartilages | Antebrachial fascia and crest of humerus | Shoulder adduction |
| *M. pectoralis transversus* | PCT | Manubrium, sternum and costal cartilages | Antebrachial fascia and crest of humerus | Shoulder adduction |
| *M. pectoralis ascendens* | PCA | Sternum and costal cartilages | Humerus, medial lesser tubercle and cranial greater tubercle with *subclavius* | Thorax support, forelimb retraction. |
| *M. subclavius* | SU | Sternum and costal cartilages | Proximal humerus with *pectoralis ascendens*, and likely dorsal scapula via fasciae | Thorax support, forelimb retraction. |
| *Mm. serrati ventrales* | SV | See *m. serratus ventralis thoracis* and *m. serratus ventralis cervicis* | Medial aspect of the scapula, proximal half | See *m. serratus ventralis thoracis* and *m. serratus ventralis cervicis* |
| *M. serratus ventralis thoracis* | SVT | Distal aspect of the first ribs | Medial aspect of the scapula, proximal half | Supports the thorax between the forelimbs |
| *M. serratus ventralis cervicis* | SVC | Transverse processes of cervical vertebrae | Medial aspect of the scapula, proximal half | Supports the head and neck between the forelimbs |
| *M. trapezius* | TP | Nuchal ligament, thoracic vertebrae 1 to 12, dorsal aspect of the ribs | Caudo-proximal part of the scapular spine | Forelimb abduction |
| *Mm. rhomboidei* | RHB | Nuchal and dorsoscapular ligaments | Scapular cartilage, medial aspect | Forelimb abduction, neck extension |
| *M. latissimus dorsi* | LD | Thoracolumbar fascia, and overall large portion of the dorsal rib cage | *Teres major* tuberosity, merging with *teres major* | Forelimb retraction |
| *M. supraspinatus* | SSP | Supraspinous fossa | Summit of the greater tubercle, above the infraspinatus insertion | Shoulder extension |
| *M. infraspinatus* | ISP | Infraspinous fossa and dorsal tip of the scapular tuberosity | Greater tubercle, caudodistal to *supraspinatus* insertion | Shoulder abduction, stabilization and extension |
| *M. subscapularis* | SSC | Medial aspect of the scapula, distal half | Lesser tubercle, likely the convexity, and articular capsule of the shoulder | Shoulder adduction |
| *M. deltoideus* | DL DLS DLA | *Pars scapularis*: Tuberosity of the scapular spine + fascia over *infraspinatus* *Pars acromialis*: distal end of scapular spine | Deltoid tuberosity of the humerus | Shoulder abduction, and shoulder flexion when combined with *teres major* |
| *M. teres major* | TRM | Medial aspect of the scapula, proximo-caudal border | *Teres major* tuberosity, merging with the *latissimus dorsi* | Shoulder adduction and internal rotation, and shoulder flexion when combined with *deltoideus* |
| *M. coracobrachialis* | CB | Coracoid process of the scapula: medial aspect, cranio-distal angle | Cranio-medial humerus, close to *brachiocephalicus* and *omotranversarius* | Shoulder adduction and internal rotation |
| *M. biceps brachii* | BB | Supraglenoid tubercle of the scapula | Medial aspect of the proximal epiphysis of the radius (radial tuberosity) | Elbow and shoulder flexion |

(Continued)

**Table 2** (*continued*)

| Name | Abb. | Origin | Insertion | Action |
|---|---|---|---|---|
| *M. brachialis* | BR | Humeral neck, extending cranio-distally | Distal to that of *biceps brachii* | Elbow flexion |
| *M. triceps brachii caput longum* | TLo | Elongated origin on the whole caudal border of the scapula | Olecranon, with a common tendon for the whole *triceps* | Elbow and shoulder extension |
| *M. triceps brachii caput laterale* | TLa | Tricipital line of the humerus | Olecranon, with a common tendon for the whole *triceps* | Elbow extension |
| *M. triceps brachii caput mediale* | TM | Caudo-medial part of the humeral diaphysis, caudal to the tuberosity of *teres major.* | Olecranon, with a common tendon for the whole *triceps* | Elbow extension |
| *M. anconeus*[1] | AN | Distal medial humeral shaft, just above the olecranon fossa | Lateral side of the olecranon | Elbow extension; accessory to the *triceps* |
| *M. tensor fasciae antebrachii* | TFA | Elongated origin on the caudal border of the scapula | Antebrachial fasciae and caudal surface of the olecranon | Elbow extension |
| *M. brachioradialis* | BRA | Proximomedial humerus, below the neck | Craniomedial radius, distal to that of the brachialis | Forearm supination |
| *M. extensor carpi radialis* | ECR | Humerus, epicondylar crest | Dorsal aspect of proximal MCIII + small tendon on MCII | Wrist extension |
| *M. ulnaris lateralis* | UL | Summit of the lateral epicondyle of the humerus | Pisiform bone, and maybe base of the plantar aspect of the MCIV | Wrist flexion |
| *M. extensor carpi obliquus* | ECO | Craniolateral surface of radius | Proximal part of dorsal MCII | Weak wrist extension |
| *M. extensor digitorum communis* | EDC | Above the radial fossa of the humerus, and lateral aspect of the radial head (*C. simum* only) | Dorsal surface of each distal phalanx | Metacarpo/interphalangeal joints extension |
| *M. extensor digitorum lateralis* | EDLaF | Lateral condyle of the humerus, craniolateral aspect, and proximo-lateral radius and ulna | Dorsal aspect of the proximal phalanx of digit IV | Digit IV joints extension |
| *M. flexor carpi radialis* | FCR | Medial epicondyle of the humerus, medial aspect, cranial to that of *FCU* | Proximo-plantar part of MCII and MCIII | Wrist flexion |
| *M. flexor carpi ulnaris* | FCU | Ulnar head: Olecranon, medial to the triceps / Humeral head: medial epicondyle, between the origins of *FDP* and *FCR* | Pisiform bone, palmar aspect | Wrist flexion |
| *M. flexor digitorum superficialis* | FDSF | Medial epicondyle of the humerus, caudo-medial aspect; most caudal origin of the four flexors | Second phalanx of all three digits, plantar aspect | Metacarpo/interphalangeal joints flexion |
| *M. flexor digitorum profundus* | FDPF | Humeral head: medial epicondyle of the humerus, medial aspect, between *FDS* and *FCU* / Ulnar head: medial olecranon | Distal phalanx of all three digits, plantar aspect | Metacarpo/interphalangeal joints flexion |

**Notes:**
[1] Muscle found only in the neonate *R. unicornis*.
Abb.: abbreviation

### Extrinsic muscles of the forelimb

The *omotransversarius* (OT) ran very close to the *brachiocephalicus* (BC) down the neck, before inserting proximal to it on the humerus (Figs. 1B, 4), with an apparent insertion on

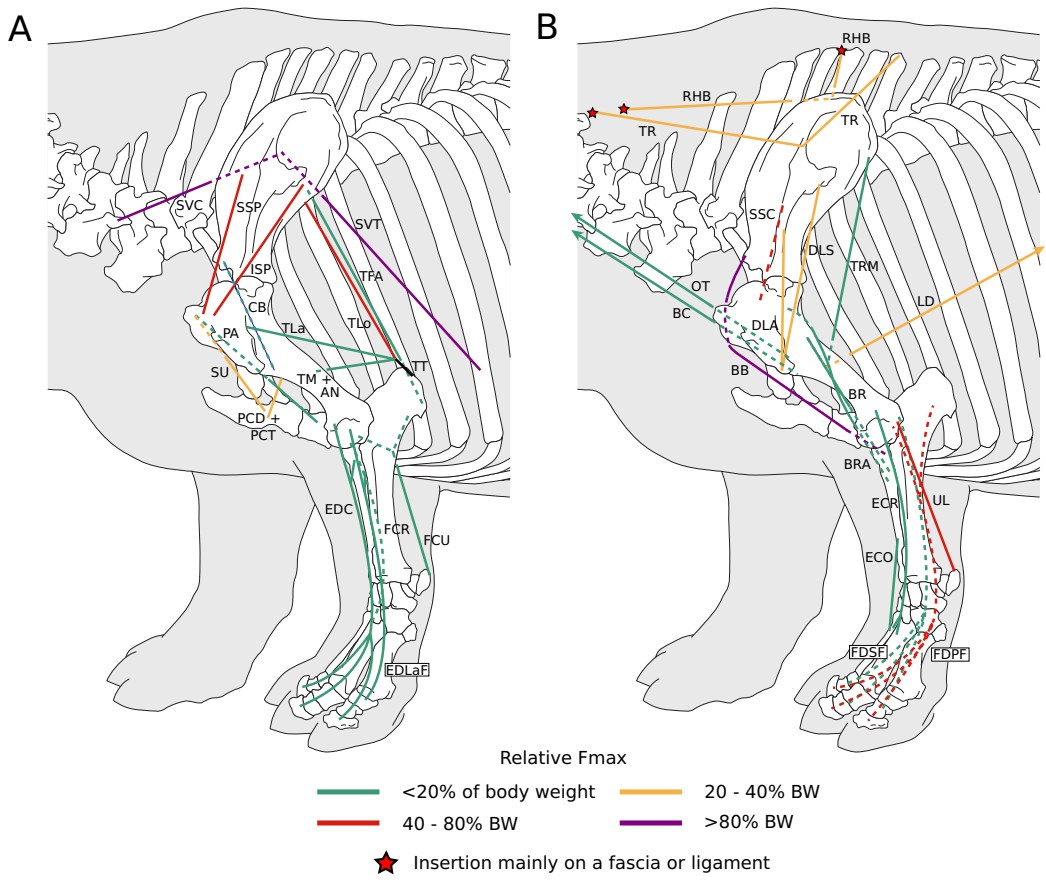

**Figure 1 Diagram representing the muscles of the left forelimb and their origins and insertions, lateral view.** Normalized Fmax values are from our adult *R. unicornis* individual; muscles whose Fmax could not be determined (*brachialis*, *extensor carpi radialis*, *flexor carpi ulnaris*) are classified as below 20% of body weight. The skeleton image is that of *R. sondaicus* (Based on *Pales & Garcia, 1981*), and is courtesy of https://www.archeozoo.org/archeozootheque/ and Michel Coutureau, under CC BY-NC-SA 4.0 license. Dashed lines represent muscles hidden by bones in lateral view. Please note that origins and insertions are not restricted to the exact points indicated by the lines. (A) *serrati ventrales thoracis* (SVT) and *cervicis* (SVC), *supraspinatus* (SSP), *infraspinatus* (ISP), *pectorales ascendens* (PA), *descendens* and *transversus* (PCD + PCT), *subclavius* (SU), *coracobrachialis* (CB), *triceps brachii caput longum* (TLo), *laterale* (TL) and *mediale* with *anconeus* (TM + AN), tendon of the *triceps brachii* (TT), *tensor fasciae antebrachiae* (TFA), *extensor digitorum communis* (EDC) and *lateralis* (EDLaF), *flexor carpi radialis* (FCR) and *ulnaris* (FCU). (B) *rhomboidei* (RHB), *trapezius* (TP), *omotransversarius* (OT), *brachiocephalicus* (BC), *subscapularis* (SSC), *deltoideus acromialis* (DLA) and *scapularis* (DLS), *latissimus dorsi* (LD), *teres major* (TRM), *biceps brachii* (BB), *brachialis* (BR), *brachioradialis* (BRA), *extensor carpi radialis* (ECR) and *obliquus* (ECO), *ulnaris lateralis* (UL), *flexor digitorum superficialis* (FDSF) and *profundus* (FDSP).

the distal scapular spine via an aponeurosis. This was already described by *Haughton (1867)* in *R. unicornis*, and distinguishes rhinoceroses from most other ungulates and elephants. In the other perissodactyls however, the muscle's aponeurosis goes from the scapular spine to the humeral crest (*Windle & Parsons, 1902*; *Bressou, 1961*; *Fisher, Scott & Naples, 2007*; *Barone, 2010*). The muscle's diameter was constant across its length, unlike in equids where it presents a triangular shape. The *brachiocephalicus* inserted at the proximal humeral crest, and tended to fuse partially with the *coracobrachialis* (CB) and

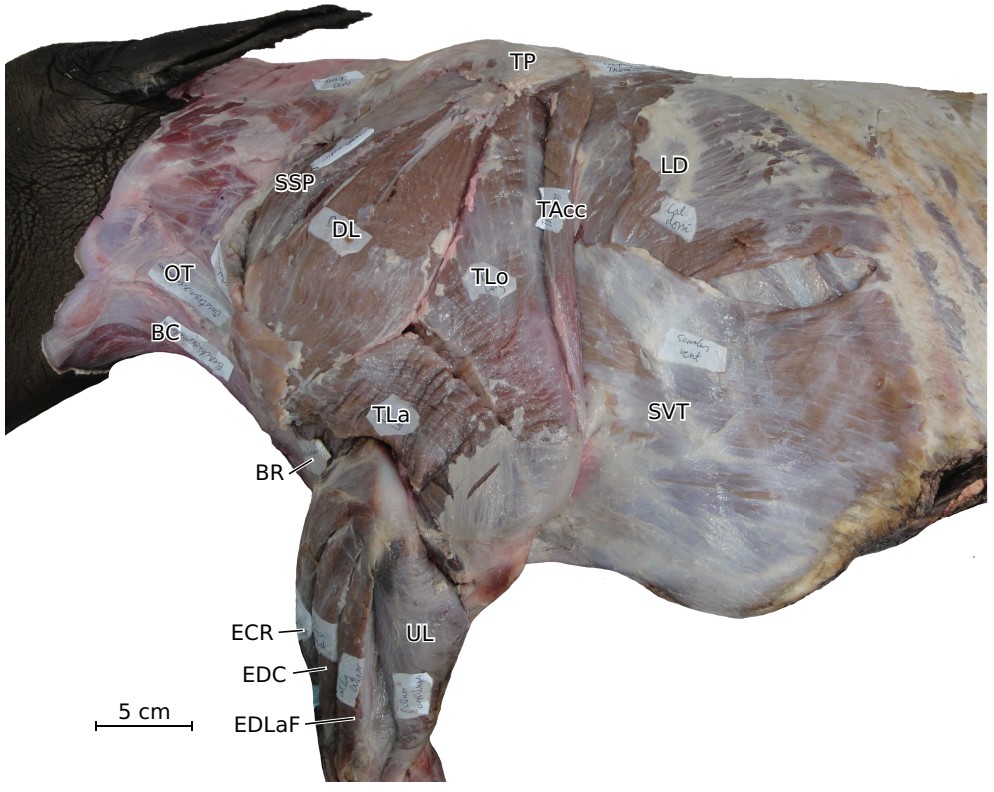

**Figure 2 Photograph of the dissection of the superficial muscles of the left forelimb (lateral view) of the neonate individual of *C. simum*, with muscle labels.** Legend as in Fig. 1, except DL: *deltoideus* and TAcc: *triceps brachii caput accessorius*.

the *omotransversarius* in the neonate *R. unicornis* when inserting; this fusion was not observed in the other specimens. It is composed of one head only, unlike what is generally observed in artiodactyls and in elephants but similar to other perissodactyls (*Miall & Greenwood, 1878*; *Campbell, 1936*; *Fisher, Scott & Naples, 2007*; *Barone, 2010*).

In our rhinoceroses, the *pectorales superficiales* (*transversus* and *descendens*, PCD and PCT) inserted next to the *brachiocephalicus* (BC) on the humeral crest (Figs. 1A, 4), like in other ungulates and in elephants (*Miall & Greenwood, 1878*; *Campbell, 1936*; *Fisher, Scott & Naples, 2007*; *Barone, 2010*; *Trenkwalder, 2013*). Contrary to horses, their insertions do not merge with that of the *brachiocephalicus*. In hippopotamuses, the *pectoralis descendens* and *transversus* are entirely fused and cannot be separated; this is not the case in rhinoceroses. The origins of the *subclavius* (SU) and of the *pectoralis ascendens* (PCA) are also like those of other ungulates and elephants. Unlike in those species however, those muscles merge before inserting on the humerus. This means that the *subclavius's* main insertion is on the proximal humerus, and not on the scapula as in other species of large ungulates and in elephants (Fig. 4). The *subclavius* may still have attached to the scapula through fascia in our rhinos, although this was difficult to determine. In horses, *Payne, Veenman & Wilson (2005)* reported an insertion of the *subclavius* on the

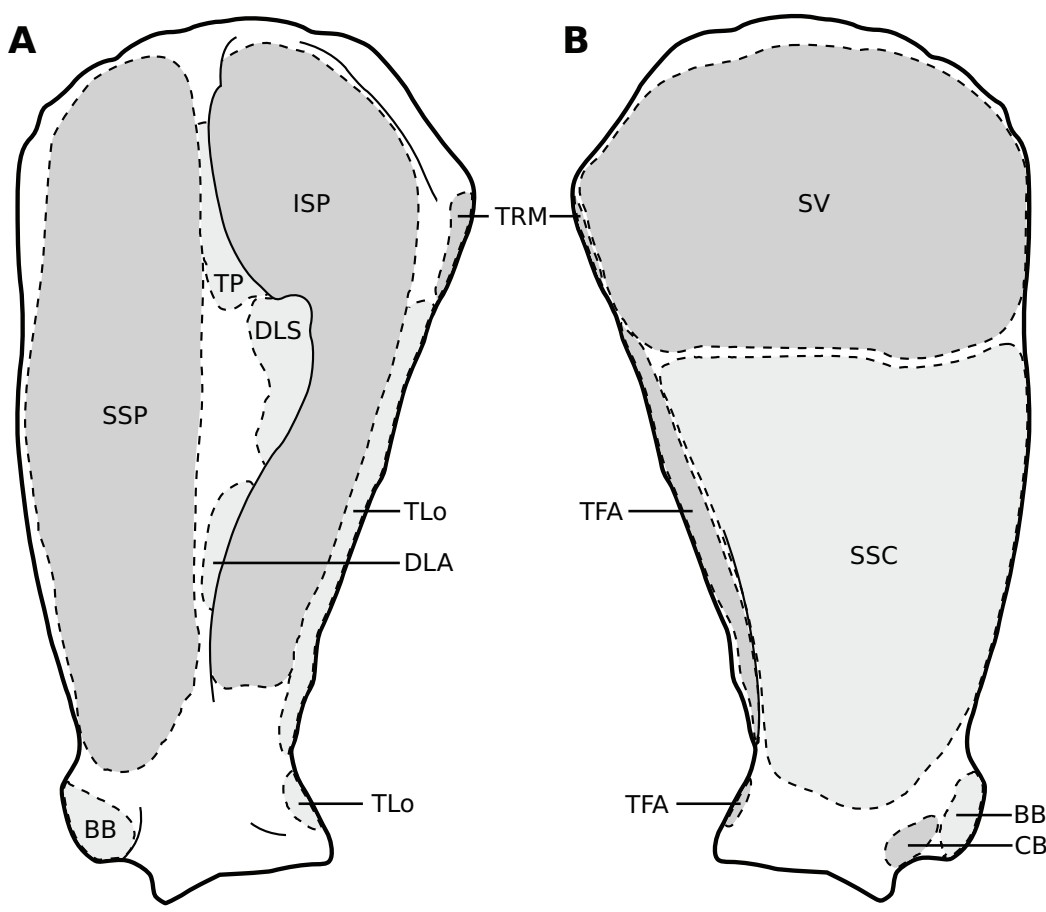

**Figure 3 Muscular origins and insertions on the scapula of rhinoceroses.** (A) Lateral view. (B) Medial view. Muscle acronyms are in Table 2. This particular scapula comes from our adult *C. simum*, but the insertions are applicable to both species.

greater tubercle, but *Barone (2010)* mentioned only the scapula, similar to tapirs (*Campbell, 1936*; *Bressou, 1961*).

The *serrati ventrales* (SVC and SVT) of rhinoceroses do not differ qualitatively from other ungulates and elephants, nor does the *latissimus dorsi* (LD), which ran along the *teres major* (TRM) as a thin tendon and inserted with it onto the *teres major* tuberosity (*Murie, 1871*; *Miall & Greenwood, 1878*; *Eales, 1928*; *Campbell, 1936*; *Bressou, 1961*; *Fisher, Scott & Naples, 2007*; *Barone, 2010*). The *trapezius* (TP) could only be separated into a *pars cervicis* and a *pars thoracis* in the neonate *C. simum*, both parts were inseparable in the other specimens. The *rhomboideus* (RHB) is similar to what is observed in other perissodactyls and large ungulates, but in elephants the *rhomboideus* is divided into several parts, due perhaps to their phylogenetic distance from the others (*Trenkwalder, 2013*).

*Muscles of the shoulder*
Like in elephants, *Hippopotamus*, suids, and *Dicerorhinus*, the *supraspinatus* (SSP) presented only one insertion in our rhinos, on the greater tubercle (Fig. 4), although *Hippopotamus* may present a minority of fibres inserting on the lesser tubercle as well,

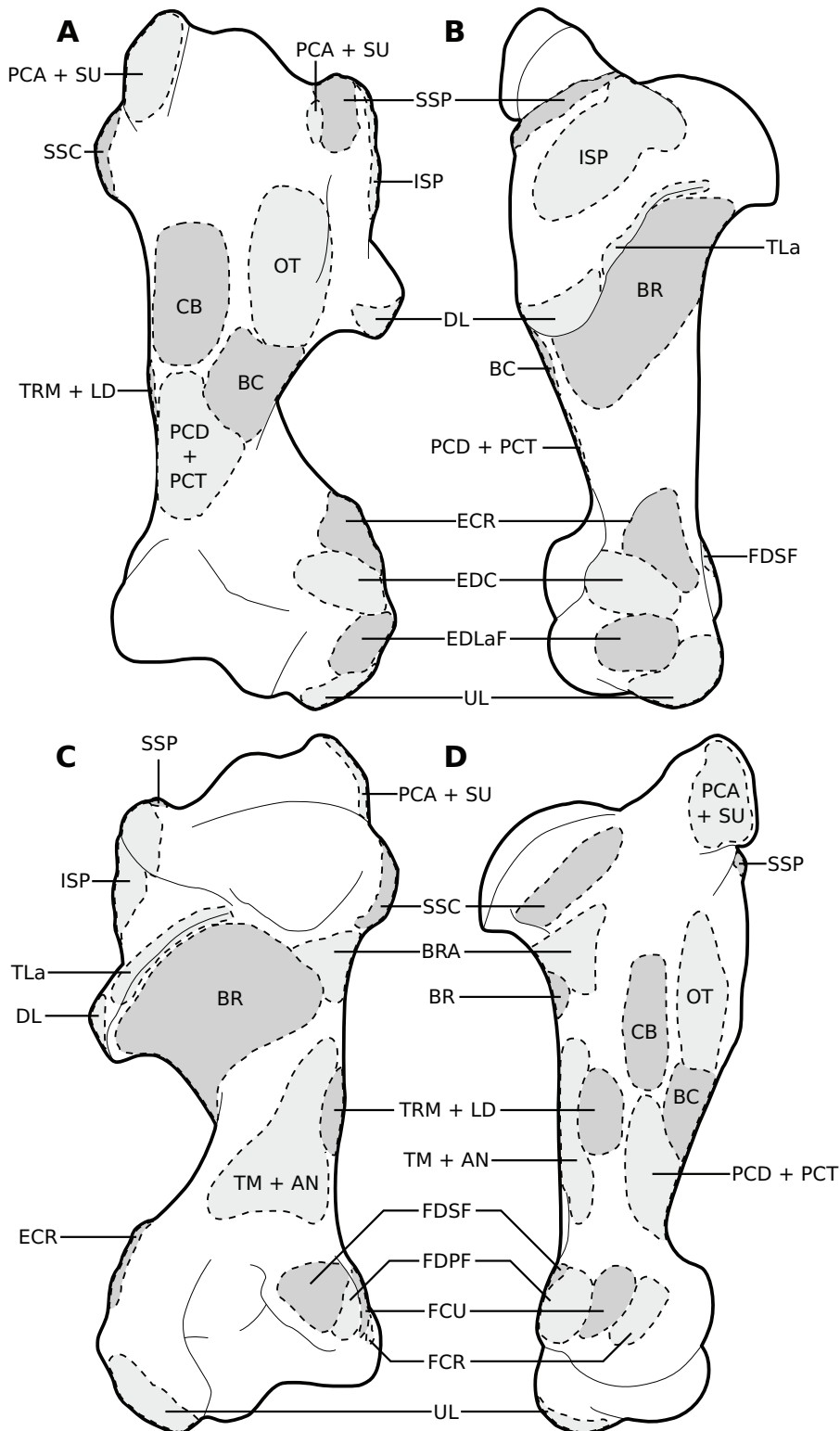

**Figure 4 Muscular origins and insertions on the humerus of rhinoceroses.** (A) Cranial view. (B) Lateral view. (C) Caudal view. (D) Medial view. Muscle acronyms are in Table 2. This particular humerus comes from our adult *C. simum*, but the insertions are applicable to both species.

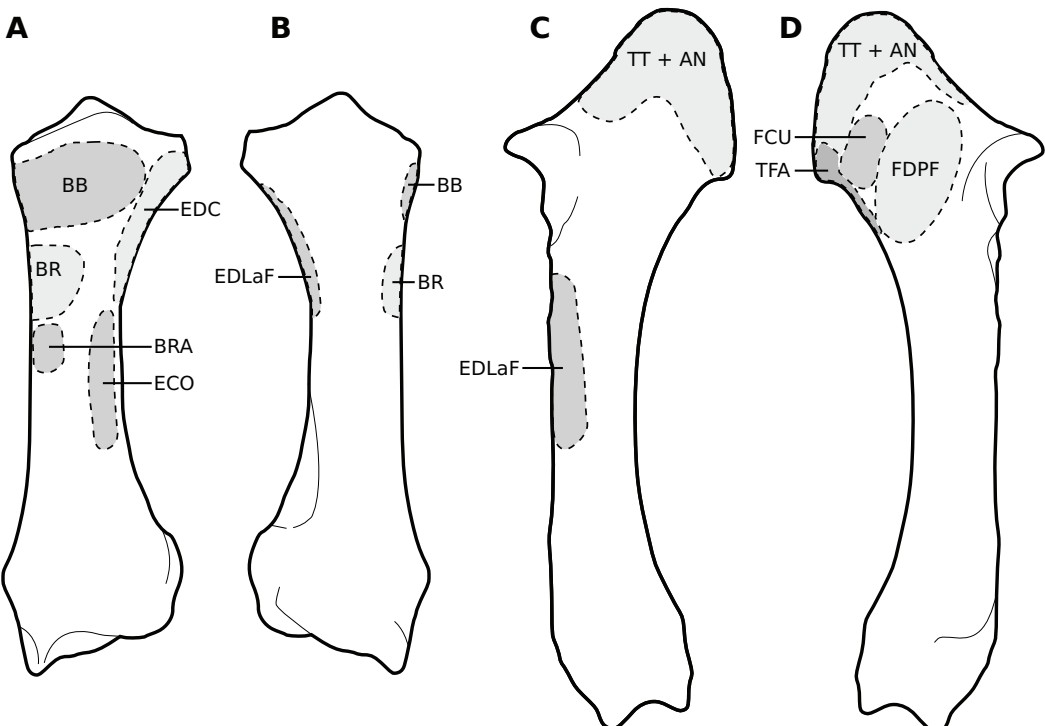

**Figure 5 Muscular origins and insertions on the radius and ulna of rhinoceroses.** (A) Radius in cranial view. (B) Radius in caudal view. (C) Ulna in lateral view. (D) Ulna in medial view. The bones are shown to the same scale. The radial origin of the *extensor digitorum communis* was not evident in our *R. unicornis* specimens. Muscle acronyms are in Table 2; TT: tendon of the *triceps brachii*. These bones come from our adult *C. simum*, but the insertions are applicable to both species.

depending on the studies. A second insertion is observed on the lesser tubercle in horses and tapirs, as well as in bovids (*Gratiolet & Alix, 1867*; *Miall & Greenwood, 1878*; *Beddard & Treves, 1889*; *Eales, 1928*; *Campbell, 1936*; *Fisher, Scott & Naples, 2007*; *Barone, 2010*; *Trenkwalder, 2013*; *MacLaren & McHorse, 2020*). It is to be noted that giraffes also present a unique insertion of the *supraspinatus* (C. Basu, 2021, personal communication). Unlike what is observed in horses and bovids, the *infraspinatus*'s (ISP) insertion on the greater tuberosity is not separable in two parts; apart from this, the muscle does not differ from what is observed in other perissodactyls, large bovids, hippopotamuses and elephants.

Unlike results reported by *Haughton (1867)*, we found two distinct parts of the *deltoideus* (DL), in the adults of both species: the *pars acromialis* (DLA) and *pars scapularis* (DLS). This is similar to what is observed in elephants, bovids, and *Choeropsis* (*Eales, 1928*; *Campbell, 1936*; *Fisher, Scott & Naples, 2007*; *Barone, 2010*; *Trenkwalder, 2013*). In *Hippopotamus*, *Gratiolet & Alix (1867)* reported that the *deltoideus* is not divided into those two parts. This division was not reported in a juvenile tapir by *MacLaren & McHorse (2020)*, but it was by *Bressou (1961)*; it may serve to provide finer control on the directions of the forces exerted by the muscle. Notably, the *pars acromialis* inserts quite proximally on the scapular spine in rhinoceroses, close to the *pars scapularis* (Fig. 3A);

this may be because the acromion is absent on the scapula of rhinoceroses (*Guérin, 1980*). Alternatively, because the muscle inserts more proximally on the spine this may have reduced the forces exerted on the acromion and allowed its eventual reduction.

In our rhino specimens, the *subscapularis* (SSC) was single-headed and mixed with fibrous fibres, as in horses. The muscle does not seem to differ much from that in other large ungulates and elephants, except hippopotamuses and domestic bovids, in which the muscle is partially split into two or more parts (*Miall & Greenwood, 1878*; *Eales, 1928*; *Campbell, 1936*; *Fisher, Scott & Naples, 2007*; *Barone, 2010*; *Trenkwalder, 2013*; *MacLaren & McHorse, 2020*). The *teres major* (TRM) is similar to that of other perissodactyls or large ungulates (*Campbell, 1936*; *Fisher, Scott & Naples, 2007*; *Barone, 2010*; *Trenkwalder, 2013*; *MacLaren & McHorse, 2020*). The *teres minor* was not found; it is possible it merged with the *infraspinatus* (ISP) of which can be deemed an accessory muscle. *Miall & Greenwood (1878)*, *Eales (1928)* and *Fisher, Scott & Naples (2007)* did report that the *teres minor* tends to blend with the *infraspinatus* in elephants and *Choeropsis*. Neither *Haughton (1867)* nor *Beddard & Treves (1889)* reported a *teres minor* in rhinoceroses, which is consistent with our hypothesis.

We observed that the *coracobrachialis* (CB) was split in two parts in our specimens, cranial and caudal, as in equids and bovids (*Barone, 2010*), inserting close to one another on the craniomedial humerus. *Bressou (1961)* also reported an incomplete division in the tapir, but other studies did not (*Murie, 1871*; *Campbell, 1936*; *MacLaren & McHorse, 2020*). *Trenkwalder (2013)* mentioned an insertion in two parts in *Loxodonta*, but *Eales (1928)* stated that the muscle is in one part, and *Miall & Greenwood (1878)* did not report subdivisions in *Elephas*, either. Only *Trenkwalder (2013)* studied an adult specimen, whereas the latter two studies were respectively of a foetus and a juvenile, so the subdivision of the muscles may have been yet to develop, as in our neonate specimens. This division is not reported in Hippopotamidae, nor, interestingly, in *Dicerorhinus* (*Gratiolet & Alix, 1867*; *Beddard & Treves, 1889*; *Campbell, 1936*; *Macdonald et al., 1985*; *Fisher, Scott & Naples, 2007*).

*Muscles of the arm*
In our specimens, the *biceps brachii* (BB) presented only one head, as in most mammals, and inserted on the radial tuberosity via a flat, very thick tendon (Figs. 1B, 5; *Barone, 2010*). In tapirs the insertion is on both the proximomedial radial head and medial coronoid process of the ulna (*Murie, 1871*; *Bressou, 1961*; *MacLaren & McHorse, 2020*). In elephants, it has been noted as originating on the articular capsule rather than the coracoid process, and inserting generally on the ulna and sometimes on the radius (*Miall & Greenwood, 1878*; *Eales, 1928*; *Trenkwalder, 2013*). The *brachialis* (BR) is like that of other perissodactyls, large ungulates and elephants, although it sometimes inserts on the ulna rather than the radius, which does not fundamentally change the muscle's action. (*Gratiolet & Alix, 1867*; *Miall & Greenwood, 1878*; *Eales, 1928*; *Campbell, 1936*; *Fisher, Scott & Naples, 2007*; *Trenkwalder, 2013*; *MacLaren & McHorse, 2020*).

The *triceps brachii* consisted of three heads (*longus, mediale, laterale*; TLo, TLa, TM); an accessory head was also observed only in the neonate *C. simum*, caudal to the long head

(Fig. 2), although this may actually correspond to the *tensor fasciae antebrachiae* (TFA). The *caput longum* and *caput laterale* of the *triceps* are similar to those observed in other perissodactyls or large ungulates and elephants. The *caput longum* was partially divided into a cranial and caudal head in the adult specimens, this is reminiscent of what has sometimes been reported in tapirs and hippopotamuses (*Campbell, 1936*; *Bressou, 1961*); the accessory head observed in the neonate *C. simum* may also correspond to the caudal of those heads. The *caput mediale* seemed to merge with the *anconeus* (AN) in all our specimens except our neonate *R. unicornis*; this has also sometimes been reported in tapirs and *Choeropsis* (*Campbell, 1936*; *Fisher, Scott & Naples, 2007*). The *caput longum* is by far the strongest one in rhinos (see *Quantitative characterisation*), followed by the *caput laterale* and then the *caput mediale*, the same pattern has been observed in horses, tapirs, elephants and most ungulates (*Miall & Greenwood, 1878*; *Eales, 1928*; *Watson & Wilson, 2007*; *Barone, 2010*; *MacLaren & McHorse, 2020*). Like in horses, the *tensor fasciae antebrachiae* originates and inserts close to the *triceps caput longum* (*Barone, 2010*). This is similar to what *Eales (1928)* and *Trenkwalder (2013)* reported in *Loxodonta*; other studies did not report this muscle.

*Muscles of the forearm*
We observed a *brachioradialis* (BRA) in three of our specimens, the neonate *C. simum* being the only exception; this is unusual in large ungulates. It is however present in tapirs as well as in elephants and sometimes in *Hippopotamus* (*Miall & Greenwood, 1878*; *Eales, 1928*; *Campbell, 1936*; *Fisher, Scott & Naples, 2007*; *Barone, 2010*; *Trenkwalder, 2013*; *Nagel et al., 2018*; *MacLaren & McHorse, 2020*). The muscle is particularly proximal in rhinos (Figs. 1B, 4, 5), originating and inserting very close to the *brachialis* (BR), to the point that both muscles may have merged in the adult *R. unicornis*.

The *extensor carpi radialis* (ECR) and *extensor carpi obliquus* (ECO) did not differ qualitatively from what is observed in other extant ungulates. The latter, which consists of the fusion of the *abductor pollicis longus* and *extensor pollicis brevis*, is particularly gracile, as usual in ungulates; it was however noted to be "strong" in *Loxodonta* (*Nagel et al., 2018*). For the *ulnaris lateralis* (or *extensor carpi ulnaris*, UL), we observed a caudal path and an insertion on the pisiform bone, meaning that this muscle clearly acts as a flexor of the carpus in both studied species (Fig. 1B). This is also observed in large artiodactyls and in equids, but not in tapirs, in which the muscle acts as an extensor due to its insertion on the fifth metacarpal (*Fisher, Scott & Naples, 2007*; *Barone, 2010*; *MacLaren & McHorse, 2020*). In adult rhinos it is the strongest muscle of the forearm (see *Quantitative characterisation*); this is in accordance with that which was found in tapirs and horses; it appears to be weak in *Choeropsis* (*Haughton, 1867*; *Brown et al., 2003*; *Fisher, Scott & Naples, 2007*). In both species, the *extensor digitorum communis*'s (EDC) main origin was on the humerus, above the radial fossa. It also presented a small radial head in our *C. simum* specimens, as in horses and *Dicerorhinus*, although it extends distally on the ulna in the latter. Our two *R. unicornis* specimens, along with hippopotamuses and elephants, seem to lack this radial head; some studies reported it *Tapirus terrestris* and *T. indicus*, others did not in the same species (*Murie, 1871*;

*Beddard & Treves, 1889*; *Windle & Parsons, 1902*; *Campbell, 1936*; *Fisher, Scott & Naples, 2007*; *Barone, 2010*; *Pereira et al., 2017*; *Nagel et al., 2018*; *MacLaren & McHorse, 2020*). This radial head might correspond to the *extensor pollicis longus*, as suggested by *Bressou (1961)* in tapirs. Given its small size, its presence or absence is most likely the result of evolutionary variation rather than a functional constraint. The *extensor digitorum lateralis*'s (EDLaF) main origin was clearly on the lateral humeral condyle, similar to that observed in most ungulates, including tapirs but not equids, where the origin is exclusively in the lateral shaft of the radius-ulna (*Beddard & Treves, 1889*; *Campbell, 1936*; *Barone, 2010*; *Nagel et al., 2018*; *MacLaren & McHorse, 2020*). The muscular belly still attached on the lateral radius and ulna while passing down the forearm.

The humeral origins of the four flexors of the carpus and digits were difficult to differentiate, but anatomical observations were consistent with the pattern known for other perissodactyls (Fig. 4; *Campbell, 1936*; *Bressou, 1961*; *Barone, 1999*, *2010*; *MacLaren & McHorse, 2020*). The *flexor carpi ulnaris* (FCU) was not found at all in the adult *R. unicornis*, whereas in the neonate it was closely appressed to the *flexor digitorum profundus* (FDPF), with which the *flexor carpi ulnaris* might have merged, as their origins on both the humerus and the ulna are close (Figs. 1, 4, 5). This muscle does not differ further from what is observed in other perissodactyls, large ungulates and elephants (*Beddard & Treves, 1889*; *Fisher, Scott & Naples, 2007*; *Barone, 2010*; *Nagel et al., 2018*; *MacLaren & McHorse, 2020*). The *flexor carpi radialis* (FCR) is similar in rhinos to that generally observed in large ungulates and elephants, and it is particularly weak, as in horses and tapirs (*Brown et al., 2003*; *MacLaren & McHorse, 2020*). In adults, the *flexor digitorum profundus* of the forelimb presented two heads, one humeral and one ulnar, separated until the tendon, where they merged with the tendon of the *superficialis* (FDSF) in our adult *C. simum* only. *Haughton (1867)* reported the same fusion in what was likely a specimen of *R. unicornis*, which means that these muscles could present a degree of variation in rhinoceroses. The *flexor digitorum profundus* is highly variable in mammals: the radial head observed in tapirs and equids was here absent or greatly reduced. *Beddard & Treves (1889)* noted only a humeral head in *Dicerorhinus*. *Hippopotamus* seems to present a radial, an ulnar and two humeral heads, *Loxodonta* an ulnar and two humeral heads, and *Elephas* only one or several humeral heads (*Miall & Greenwood, 1878*; *Campbell, 1936*; *Barone, 2010*; *Nagel et al., 2018*; *MacLaren & McHorse, 2020*).

### Hindlimb

The anatomy of each muscle of the hindlimb was recorded (Table 3, Figs. 6, 7), and their origin and insertion on the bones were determined (Figs. 8, 9). As for the forelimb, several muscles were damaged before or during dissection: the *popliteus* (PP) in the adult *C. simum*, and the *obturator et gemelli* (OG) in the neonate *R. unicornis*. Others were not found at all: the *psoas minor* (PMN) in both *R. unicornis*, the *gluteus profundus* (GPF), *popliteus* and *extensor digitorum lateralis* (EDLaH) in the neonate *R. unicornis*. In the neonate *C. simum*, both *flexores digitorum* were merged and impossible to separate, as well as the two heads of the *gastrocnemius* (GC). The *piriformis*, *quadratus femoris*,

**Table 3 General origins and insertions of the muscles of the hindlimb in rhinoceroses, with their main action (anatomically estimated function, based on *Barone, 2010*).**

| Name | Abb. | Origin | Insertion | Action |
|---|---|---|---|---|
| *M. iliacus* | IL | Craniomedial surface of illium. Iliac fossa | Lesser trochanter, common with *psoas major* | Hip flexion, hip external rotation |
| *M. psoas major* | PMJ | Last ribs and thoracolumbar vertebrae, ventral surfaces | Lesser trochanter, common with *iliacus* | Hip flexion, hip external rotation, lumbar region flexion |
| *M. psoas minor* | PMN | Thoracolumbar vertebrae, ventral surfaces, medial to *psoas major* | *Psoas minor* tubercle; most fibres are continuous with the *sartorius* | Lumbar region flexion |
| *M. gluteus medius* | GMD | Wide origin along the dorsal caudal ilium | Summit of the greater trochanter, craniolateral side | Hip extension |
| *M. gluteus profundus* | GPF | Ventrocaudal part of the iliac wing | Convexity (cranial part) of the greater trochanter, medial side | Hip abduction, hip extension |
| *M. gluteus superficialis* | GSP | Caudal corner of the ilium, caudal to *gluteus medius* | Third trochanter, lateral aspect | Hip abduction |
| *Mm. obturator et gemelli* | OG | Ventral pubis and ischium | Trochanteric fossa | Hip external rotation, also hip abduction or adduction depending on the muscle |
| *M. tensor fasciae latae* | TFL | Cranio-lateral *tuber coxae*, caudal to *sartorius*, cranial to *gluteus medius* | *Fasciae latae*, around the knee | Hip flexion, knee extension |
| *M. gluteobiceps* | GB | *Biceps femoris*: Ischial tuberosity *Gluteofemoralis*: sacrosciatic ligament, dorsal ilium and sacral vertebral bodies | Tibial crest and lateral patella as a fibrous band, and the calcaneus by a caudal extension | Hip, knee and ankle extension (weakly). |
| *M. semimembranosus* | SM | Ischial tuberosity, medial to *semitendinosus* | Medial epicondyle of femur, medial patella and medial proximal tibia of tibia | Hip extension, knee flexion |
| *M. semitendinosus* | ST | One head on the sacrum and the first caudal vertebrae, one head on the ischial tuberosity, lateral to *semimembranosus* | Patella, medial tibia, and leg fasciae down to the calcaneus | Hip extension, knee flexion, ankle extension |
| *M. quadriceps femoris* | QF | See *rectus femoris, vastus medialis* and *vastus lateralis* | | |
| *M. rectus femoris* | RF | Ilium, cranial to the acetabulum | Dorsal patella | Knee extension |
| *M. vastus medialis* | VM | Medial proximal femoral shaft | Dorso-medial patella | Knee extension |
| *M. vastus lateralis* | VL | Lateral proximal femoral shaft, and a small attachment to the ventral ilium caudal to the iliac crest. | Dorso-lateral patella | Knee extension |
| *M. sartorius* | SRT | One head on the inguinal ligament, the other on the tuber coxae (*R. unicornis* only) | One head on the proximo-medial tibia, the other on the medial patella (*R. unicornis* only) | Knee adduction |
| *M. gracilis* | GRC | Pelvic symphysis | Fascia of the medial stifle and cranio-medial tibia | Hip adduction, tensor of the fasciae latae |
| *M. pectineus* | PTN | Prepubic tendon and iliopubic eminence | Distal third of the medial femur | Hip adduction, flexion and internal rotation |
| *Mm. adductores* | ADD | Ventromedial aspect of the pelvis | *Adductor brevis*: medial femur; *Adductor magnus*: medial tibial condyle and fasciae around the knee | Hip adduction |
| *M. tibialis cranialis* | TCR | Lateral tibial cotyle and tibial fossa | Medial aspect of the medial cuneiform | Ankle flexion |

(Continued)

| Name | Abb. | Origin | Insertion | Action |
|------|------|--------|-----------|--------|
| Table 3 (continued) | | | | |
| *M. fibularis tertius* | FIT | Distal cranial femur (extensor fossa) | Dorsal aspect of MT III | Auxiliary to the *tibialis cranialis* |
| *M. extensor digitorum longus* | EDLo | Distal cranial femur (extensor fossa) | Dorsal aspect of each of the distal phalanges + MTII | Digit extension, ankle flexion |
| *M. fibularis longus* | FIL | Head and shaft of the fibula and the lateral tibial cotyle | Lateral malleolus and proximal lateral MTIV | Abduction and external rotation of the ankle |
| *M. extensor digitorum lateralis* | EDLaH | Lateral aspect of the fibular head | Dorsolateral aspect of the distal phalanx of digit IV | Extension and weak abduction of digit IV |
| *M. popliteus* | PP | Lateral aspect of the lateral condyle of the femur, in a small fossa | Proximal caudal tibia | Knee flexion and internal rotation. |
| *M. gastrocnemius* | GC GCL GCM | Resp. lateral and medial supracondylar tuberosity for *caput laterale* and *caput mediale* | Cranial tuber calcanei | Ankle extension |
| *M. flexor digitorum superficialis* | FDSH | Supracondylar fossa | Plantar aspect of the proximal part of the second phalanges of all digits | Metacarpo/interphalangeal joints flexion |
| *Mm. flexores digitorum profundi* | FDPH | Caudal tibia and fibula | Plantar aspect of the distal phalanx of each digit | Metacarpo/interphalangeal joints flexion |

Note:
Abb.: abbreviation.

*articularis coxae, soleus, tibialis caudalis, extensor hallucis longus* and *fibularis brevis* were not found in any specimen.

*Muscles of the pelvis*

The *iliacus* (IL) and the *psoas major* (PMJ) are similar to what is observed in other perissodactyls and in large ungulates and elephants; they did not merge completely but inserted close to one another on the lesser trochanter (Figs. 6A, 8). The fusion of these muscles seems more prominent in *Hippopotamus* and *Bos taurus* than in perissodactyls; the degree of fusion in elephants is unclear (*Gratiolet & Alix, 1867*; *Murie, 1871*; *Miall & Greenwood, 1878*; *Eales, 1928*; *Bressou, 1961*; *Barone, 2010*; *Fisher, Scott & Adrian, 2010*). The *psoas minor* (PMN) inserted on the tuber coxae and differs from other taxa in that most of its fibres are continuous with the *sartorius* (SRT). This was already described by *Beddard & Treves (1889)* in *Dicerorhinus*, and therefore appears an apomorphy of Rhinocerotidae, although *Haughton (1867)* only noted in *Rhinoceros* that the *sartorius* originated "close" to the *psoas minor*, without further precision (see *Murie, 1871*; *Miall & Greenwood, 1878*; *Eales, 1928*; *Bressou, 1961*; *Payne et al., 2005*; *Barone, 2010*; *Fisher, Scott & Adrian, 2010*).

Three gluteal muscles were recorded: the *gluteus superficialis* (GSP), *medius* (GMD) and *profundus* (GPF; Fig. 6B); the *accessorius* was missing or merged with the *profundus*. They are in general similar to what is observed in horses and tapirs, with the exception that the *superficialis* was noted as being chiefly aponeurotic in tapirs and relatively weak in horses (*Murie, 1871*; *Payne et al., 2005*; *Barone, 2010*). *Haughton (1867)* recorded the *superficialis* as inserting on the fibula with tendinous strips for the greater and third trochanters in *R. unicornis*; we did not find such attachments. In *Hippopotamus* and it

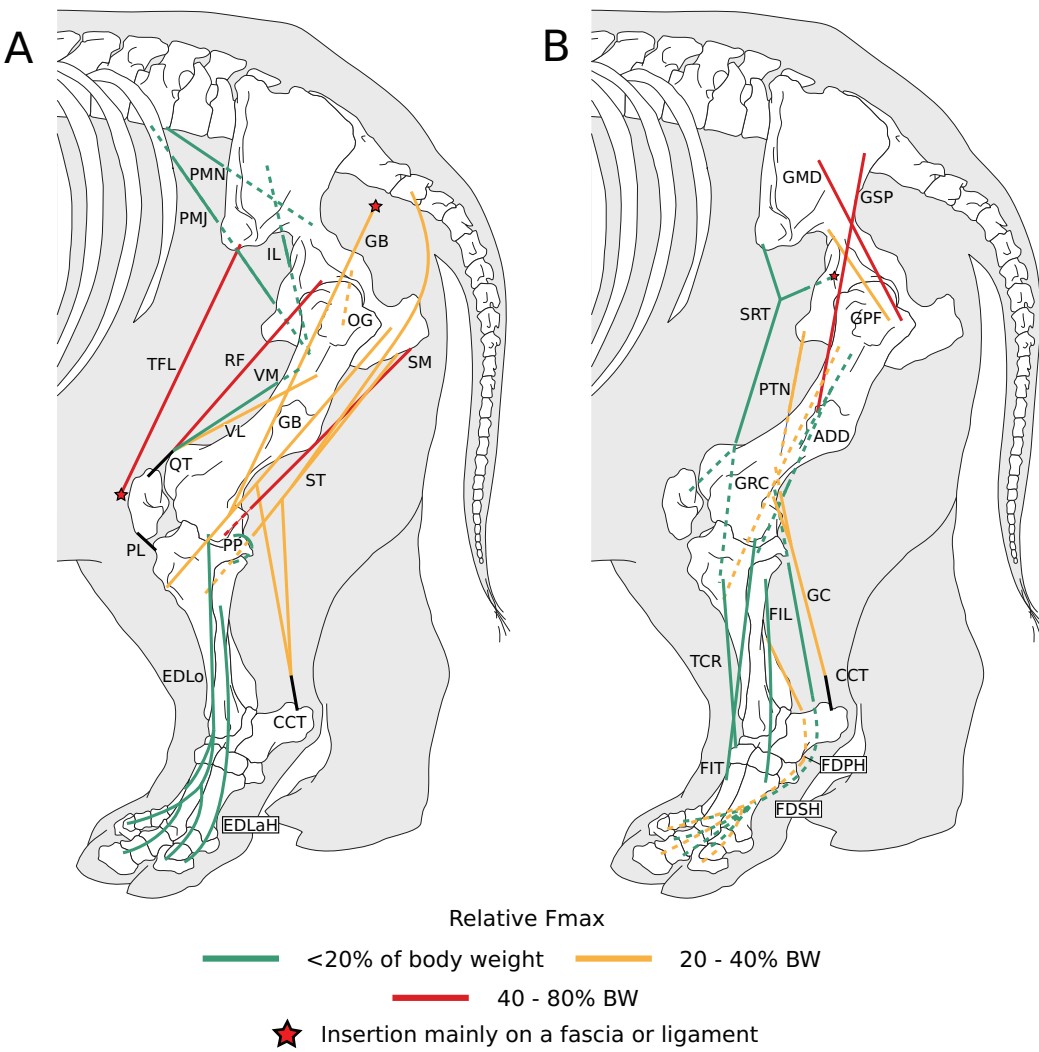

**Figure 6 Diagram representing the muscles of the left hindlimb and their origins and insertions, lateral view.** Normalized Fmax values are those of our adult *R. unicornis* individual; muscles whose Fmax could not be determined (mm. *psoas minor*, *fibularis tertius*, *fibularis longus*) are classified as below 20% of body weight. The skeleton image is that of *R. sondaicus* (based on *Pales & Garcia, 1981*), and is courtesy of https://www.archeozoo.org/archeozootheque/ and Michel Coutureau, under CC BY-NC-SA 4.0 license. The diagram is split in two to improve readability. Dashed lines represent muscles hidden by bones in lateral view. Please note that origins and insertions are not restricted to the exact points indicated by the lines. (A) *Psoas minor* (PMN), *psoas major* (PMJ), *iliacus* (IL), *obturator et gemelli* (OG), *tensor fasciae latae* (TFL), *gluteobiceps* (GB), *semimembranosus* (SM), *semitendinosus* (ST), *rectus femoris* (RF), *vastus medialis* (VM) and *lateralis* (VL), *quadriceps* tendon (QT), patellar ligaments (PL), *popliteus* (PP), *extensor digitorum longus* (EDLo) and *lateralis* (EDLaH), *common calcaneal tendon* (CCT). (B) *Gluteus superficialis* (GSP), *medius* (GMD) and *profundus* (GPF), *sartorius* (SRT), *gracilis* (GRC), *pectineus* (PTN), *adductores* (ADD), *tibialis cranialis* (TCR), *fibularis tertius* (FIT); *fibularis longus* (FIL), *gastrocnemius* (GC), *common calcaneal tendon* (CCT) and *flexor digitorum superficialis* (FDSH) and *profundus* (FDPH).

seems artiodactyls in general, the *superficialis* is merged with the *gluteobiceps*; this was not recorded here (*Barone, 2010*; *Fisher, Scott & Adrian, 2010*). The *gluteus medius* and *profundus* do not differ from what is generally observed in perissodactyls or other large ungulates.

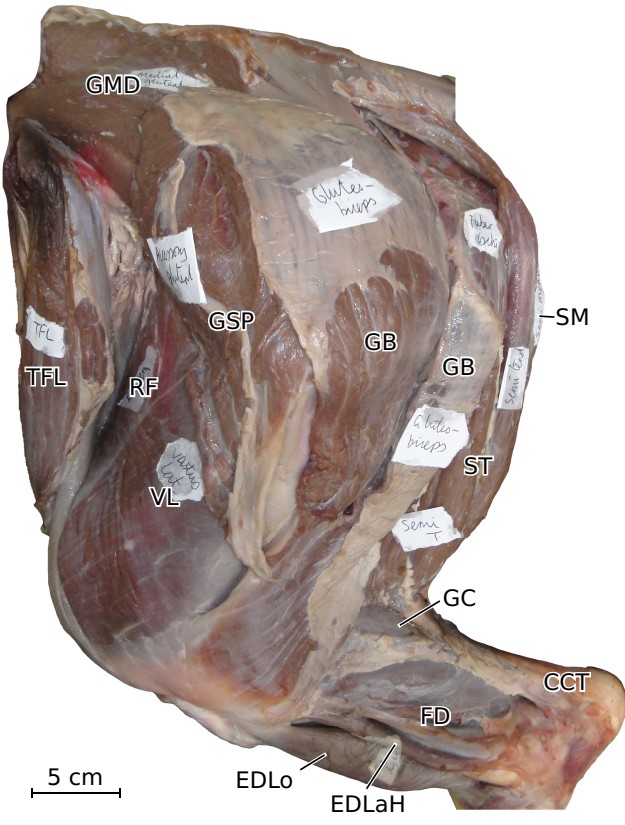

**Figure 7** **Photograph of the dissection of the superficial muscles of the left hindlimb (lateral view) of the neonate individual of *C. simum*, with muscle labels.** Legend as in Fig. 6.

The *obturator internus, obturator externus* and the *gemelli* (OG) were fused and hard to distinguish from one another, and all inserted onto the trochanteric fossa. This has not been described in perissodactyls, large ungulates or elephants, to our knowledge. This arrangement may provide more stability to the hip joint, by ensuring that the abduction or adduction functions of the different components of this muscle regulate each other. The *articularis coxae* muscle was absent in our specimens and was not reported by *Haughton (1867)* in *Rhinoceros* nor *Beddard & Treves (1889)* in *Dicerorhinus*, either. It has been reported in equids and hippopotamuses, but not in elephants, nor in most artiodactyls and in tapirs (*Haughton, 1867*; *Murie, 1871*; *Miall & Greenwood, 1878*; *Eales, 1928*; *Bressou, 1961*; *Barone, 2010*; *Fisher, Scott & Adrian, 2010*).

*Muscles of the thigh*
The *tensor fasciae latae* (TFL) formed a fibrous band around the knee, tightly bound with the *sartorius* (SRT), superficial to the *quadriceps femoris* (QF), similar to other large ungulates and elephants. It has been noted, albeit qualitatively, as being especially strong in tapirs, elephants and *Hippopotamus*, which is congruent with what we measured in

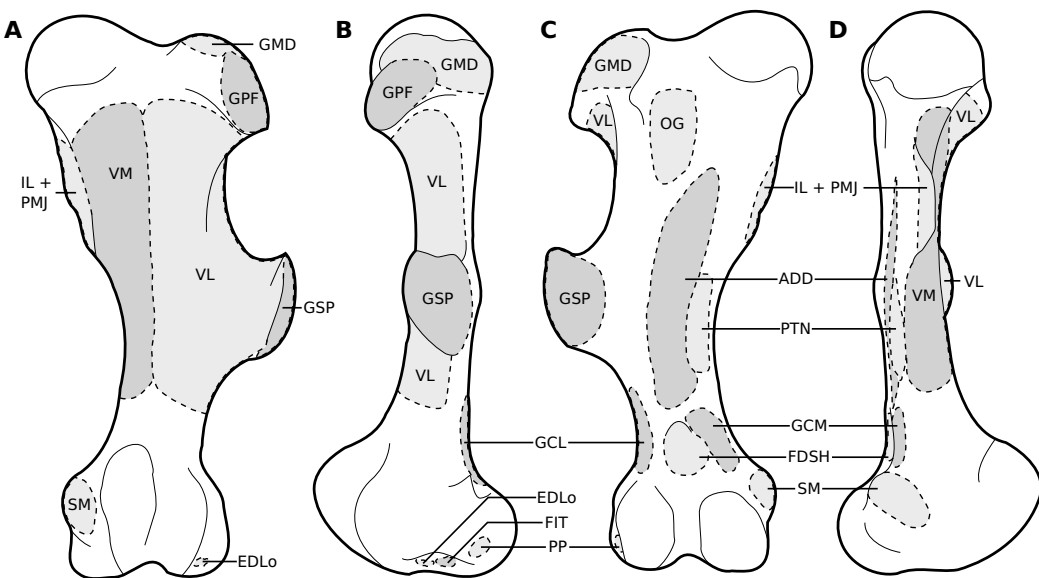

**Figure 8 Muscular origins and insertions on the femur of rhinoceroses.** (A) Cranial view (B) Lateral view. (C) Caudal view. (D) Medial view. Muscle acronyms are in Table 3. This particular femur comes from our adult *C. simum*, but the insertions are applicable to both species.

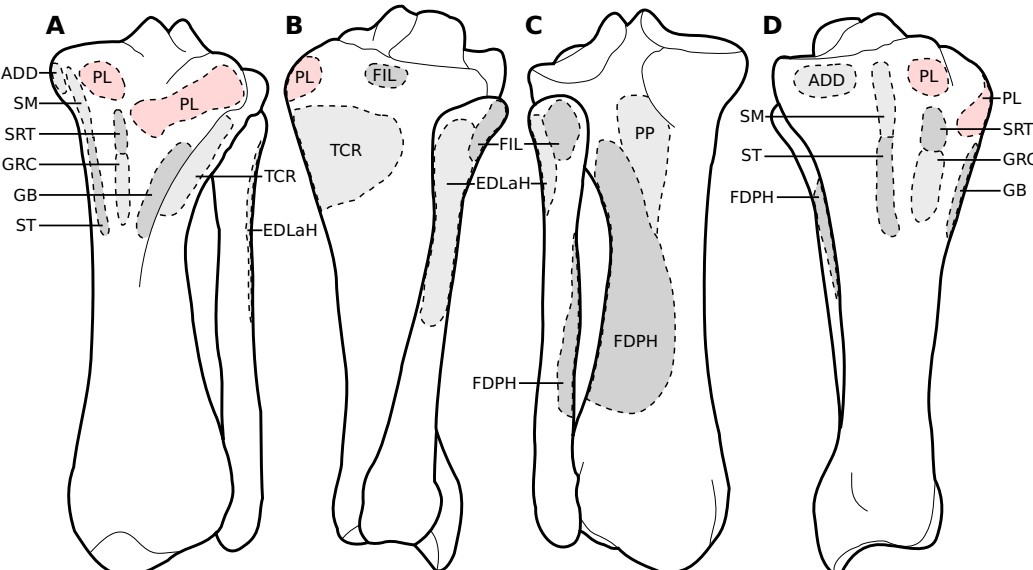

**Figure 9 Muscular origins and insertions on the tibia and fibula of rhinoceroses.** (A) Cranial view. (B) Lateral view. (C) Caudal view. (D) Medial view. The patellar ligaments (PL, in pink) are shown given their important action in transmitting the force generated by the *quadriceps femoris* on the patella. Muscle acronyms are in Table 3. These bones come from our adult *C. simum*, but the insertions are applicable to both species.

rhinos (see *Quantitative characterisation*); this strength is most likely useful for the support and propulsion of a heavy animal (*Haughton, 1867*; *Murie, 1871*; *Miall & Greenwood, 1878*; *Eales, 1928*; *Bressou, 1961*; *Barone, 2010*; *Fisher, Scott & Adrian, 2010*).

The *biceps femoris* and *gluteofemoralis* merged two thirds of the way down the femur, forming a *gluteobiceps* (GB) that inserted mainly on the lateral patella and tibia, via a fibrous band reaching up to the common calcaneal tendon (Figs. 6A, 7). The presence of a *gluteobiceps*is characteristic of numerous ungulates, although it is often simply called *biceps femoris*. In horses and tapirs, it is composed of three heads, but in rhinoceroses we only found two. In large artiodactyls, there are two heads as well, and the cranial one (the *gluteofemoralis*) merges with the *gluteus superficialis* (GSP). In elephants, *Miall & Greenwood (1878)* and *Eales (1928)* reported only one head to the *biceps femoris*; it is unclear if the *gluteofemoralis* indeed merged with it. The *semimembranosus* (SM) is like that of horses, with an insertion going from the medial epicondyle of the femur to the proximal tibia, except that in rhinos it extends further distally. This is similar to what has been reported in tapirs and domestic bovids (*Murie, 1871*; *Bressou, 1961*; *Barone, 2010*). Unlike in tapirs though, the muscle originates from only one head. *Beddard & Treves (1889)* noted a fusion with the *semitendinosus* (ST) in *Dicerorhinus*; this was not recorded here except in the neonate *R. unicornis*, although the two muscles were close in the other specimens. The *semimembranosus* appears quite different in *Hippopotamus*, where it merges with the *adductor communis* and inserts up to the crural fascia (*Fisher, Scott & Adrian, 2010*). In elephants, the origin is in two parts, and the insertion is more distal, from the proximal tibia to the malleolus and the leg fasciae (*Miall & Greenwood, 1878*; *Eales, 1928*). The *semitendinosus* is like that of the horse and tapir, with one head originating near the sacrum, the other on the ischial tuberosity; the two heads were more clearly separated in the adult *R. unicornis* than in the other specimens. The sacral head is not observed in *Hippopotamus*, domestic bovids, and *Elephas*, but *Eales (1928)* reported its presence in *Loxodonta*. The insertion is similar in all species (*Murie, 1871*; *Miall & Greenwood, 1878*; *Eales, 1928*; *Bressou, 1961*; *Barone, 2010*; *Fisher, Scott & Adrian, 2010*).

The *quadriceps* is composed of only three heads: the *rectus femoris* (RF), *vastus lateralis* (VL) and *vastus medialis* (VM). The *vastus intermedius* has been noted in horses as being split into two parallel parts that each tend to merge with the other corresponding *vastus* (*Barone, 2010*). This anatomy is likely the case in rhinoceroses as well, to a greater extent of merging that makes the *intermedius* indistinguishable in our specimens. The muscle is still distinguishable in tapirs and was reported by *Haughton (1867)* in *Rhinoceros* as well, pointing to a degree of individual variability for this muscle (*Murie, 1871*; *Bressou, 1961*). In *Dicerorhinus*, only two *vasti* are reported, and they are even reported to merge together and with the *rectus femoris* (*Beddard & Treves, 1889*). *Hippopotamus* also lacks a separate *vastus intermedius*, but elephants possess all four heads of the quadriceps. As noted in tapirs, *Hippopotamus* and elephants and contrary to horses, the *vastus lateralis* was larger than the *medialis* (*Gratiolet & Alix, 1867*; *Murie, 1871*; *Miall & Greenwood, 1878*; *Eales, 1928*; *Bressou, 1961*; *Payne et al., 2005*; *Fisher, Scott & Adrian, 2010*).

In our two specimens of *R. unicornis* the *sartorius* (SRT) consisted of two distinct heads, merging then separating in their middle section, one going from the inguinal ligament to the proximo-medial tibia, the other from the tuber coxae to the medial patella.

Only the former was found in *Ceratotherium*. This arrangement in *R. unicornis* is surprising, and reminiscent of what is observed notably in domestic carnivores, where the *sartorius* indeed originates from the tuber coxae (*Barone, 2010*). The first head was similar to the only head observed in *C. simum*, *Dicerorhinus*, horses and tapirs (*Murie, 1871*; *Beddard & Treves, 1889*; *Bressou, 1961*; *Payne et al., 2005*; *Barone, 2010*). Notably, *Haughton (1867)* also reported only one head in *R. unicornis*. The *sartorius* of domestic bovids and *Hippopotamus* is proximally divided in two. *Miall & Greenwood (1878)* reported in *Elephas* a muscle like what we observed in *C. simum* but inserting on the leg fasciae close to the proximo-medial tibia. *Eales (1928)* reported the *sartorius* as being vestigial in *Loxodonta*. This muscle seems to be particularly weak in perissodactyls, although tapirs lack quantitative data (*Murie, 1871*; *Bressou, 1961*; *Payne et al., 2005*). Unlike *Hippopotamus* and domestic bovids but similar to horses, the insertion(s) of the *sartorius* in both species are not common with the *gracilis*'s. The *gracilis* (GRC) is like that of *Dicerorhinus*, horses and tapirs in being very large and relatively flat, even though unlike in those species, it did not extend to the patella via fasciae in our species. The muscle is similar to that of other perissodactyls and elephants in its origin and insertion, except that it divides in two distally in tapirs (*Murie, 1871*; *Miall & Greenwood, 1878*; *Beddard & Treves, 1889*; *Eales, 1928*; *Barone, 2010*). In *Hippopotamus*, it is fused proximally with the *semitendinosus* and *semimembranosus* (*Fisher, Scott & Adrian, 2010*).

The *pectineus* (PTN) consisted of two heads, one larger than the other, in the adult *R. unicornis*, whereas the other specimens showed only one head. It is similar in insertion and origin to that of horses, *Dicerorhinus*, *Hippopotamus* and elephants and to that which was reported by *Bressou (1961)* in tapirs. Conversely, *Murie (1871)* reported a much more proximal insertion on the trochanteric fossa in tapirs. The two heads observed in *R. unicornis* may correspond to the proximal subdivisions of this muscle observed in horses; alternatively, one of them could correspond to the *adductor longus*, which is said to have merged with the *pectineus* in horses and was not found separately in our rhinoceroses. Unlike in horses and tapirs, the *adductor magnus* and *brevis* (ADD) are merged in their proximal part. Compared to horses, the *adductor magnus* inserts more distally on the proximal medial tibia and around the fasciae of the knee, rather than on the femur (*Murie, 1871*; *Bressou, 1961*; *Barone, 2010*). This more distal insertion is reminiscent of that of the *pectorales* in the forelimb, and likely provides the muscle with a larger lever arm to adduct and potentially retract the leg as well. This is coherent with what *Beddard & Treves (1889)* reported in *Dicerorhinus*, if their *adductor magnus* corresponds to our *brevis* and their *longus* to our *magnus*. Tapirs also present a tibial insertion of their *adductores*, although merged with the *semimembranosus* (SM; *Bressou, 1961*). In *Hippopotamus*, the *adductores* are merged, but distally, not proximally; their insertion is similar to that of rhinoceroses but the caudal part of the muscle merges with the *semimembranosus* (*Fisher, Scott & Adrian, 2010*). Elephants do not present the distal insertion observed in rhinoceroses, tapirs and *Hippopotamus*, as their *adductores* muscles insert more proximally, exclusively on the femur (*Miall & Greenwood, 1878*; *Eales, 1928*). This could be due to their proportionally much longer legs.

*Muscles of the leg*

The *tibialis cranialis's* (TCR) insertion was on the medial cuneiform in our *R. unicornis* and *C. simum*, slightly more proximal than that of *Dicerorhinus*, *Hippopotamus*, tapirs and horses, which are placed on the medial cuneiform and on the second (*Hippopotamus*, *Dicerorhinus T. indicus* in some studies) or third (*T. indicus* in other studies, horses) metatarsal (*Murie, 1871*; *Beddard & Treves, 1889*; *Bressou, 1961*; *Barone, 2010*; *Fisher, Scott & Adrian, 2010*). This is consistent with what *Haughton (1867)* reported in *R. unicornis*. In elephants, the muscle is partially merged with the *extensor digitorum longus* and may originate more distally on the tibial shaft (*Miall & Greenwood, 1878*; *Eales, 1928*; *Weissengruber & Forstenpointner, 2004*). It is weaker than the *extensor digitorum longus*, as is common in ungulates. We report here two *fibulares* muscles, the *tertius* (FIT) and the *longus* (FIL), although the *fibulares* muscles were exceedingly difficult to identify in our specimens, due to their distinct reduction. This is reminiscent of what is observed in horses, where the *fibularis tertius* is entirely tendinous and the *longus* absent (*Barone, 2010*). In tapirs, the *tertius* appears to merge with the *tibialis cranialis* (*Bressou, 1961*). The *fibulares* are well developed in *Hippopotamus* and in domestic bovids, and are also present in elephants where *Weissengruber & Forstenpointner (2004)* reported both a *longus* and a *brevis*.

The *extensor digitorum longus's* (EDLo) origin was on the extensor fossa (Fig. 8), similar to that observed in other perissodactyls and large ungulates, except in *Dicerorhinus* and in elephants where it originates on the lateral tibial condyle and even down to the tibial shaft in *Elephas* (*Murie, 1871*; *Miall & Greenwood, 1878*; *Beddard & Treves, 1889*; *Eales, 1928*; *Bressou, 1961*; *Barone, 2010*; *Fisher, Scott & Adrian, 2010*). The *extensor digitorum longus* divided into two muscular bellies distally: the medial one inserted directly around the second metatarsal, the other split into three tendons, one for each distal phalanx (Fig. 6A). The insertions seem highly variable in the taxa we compared, and the different tendons were tightly bound and hard to differentiate, so a confusion on *Haughton's (1867)* part is not excluded. *Haughton (1867)* also reported in *R. unicornis* a division in two with a medial belly inserting proximally, but on the medial cuneiform rather than on the metatarsus. The lateral belly inserted only on the proximal phalanges of digits II and IV in his specimen, whereas in our specimens, the insertion was on the distal phalanx of each finger. In *Dicerorhinus*, a simple division in three tendons, one for each toe, has been observed, as in tapirs. Equids have only one tendon, for the single digit (*Murie, 1871*; *Beddard & Treves, 1889*; *Bressou, 1961*; *Barone, 2010*). The *extensor digitorum lateralis* of the hindlimb (EDLaH) was not reported by *Haughton (1867)* nor *Beddard & Treves (1889)*. It is indeed a very gracile muscle, which may have been missing in their specimens, as in our neonate *R. unicornis*. It is gracile in equids and tapirs as well, being almost fibrous in the latter (*Bressou, 1961*; *Payne et al., 2005*). Its origin on the proximal fibula is similar to equids, tapirs, domestic bovids and *Hippopotamus*; in elephants however, the muscle also originates from the lateral collateral ligament and the tibial shaft. The insertion is similar to that of tapirs; in horses it is on the third digit as it is the only remaining digit; in *Hippopotamus* the insertion is on the distal phalanx of

digits IV and V. Additionally, in horses and *Hippopotamus* the tendons merge with that of the *extensor longus*, which was not observed here. In elephants, the insertion is more proximal, on the metatarsals and the proximal phalanges of digits IV and V (*Murie, 1871*; *Miall & Greenwood, 1878*; *Beddard & Treves, 1889*; *Eales, 1928*; *Bressou, 1961*; *Barone, 2010*; *Fisher, Scott & Adrian, 2010*).

The *gastrocnemius* (GC) does not differ qualitatively from what is observed in other perissodactyls and large ungulates, except that the lateral head (GCL) is stronger in rhinoceroses, in contrast with what was measured in horses, and qualitatively observed in *Hippopotamus* (See *Quantitative characterisation*; *Payne et al., 2005*; *Fisher, Scott & Adrian, 2010*). In elephants, the medial head (GCM) is divided in two proximally, and the origins are generally on the joint capsule rather than directly on the shaft. The *soleus* seemed to have merged with the *gastrocnemius* in our rhinos; it is reduced in the other perissodactyls and absent in *Hippopotamus*, which is consistent with our observations (*Gratiolet & Alix, 1867*; *Murie, 1871*; *Bressou, 1961*; *Payne et al., 2005*; *Fisher, Scott & Adrian, 2010*). This is in contrast with elephants where it is quite bulky (*Weissengruber & Forstenpointner, 2004*). The *popliteus* (PP) is identical to that of the other perissodactyls or large ungulates.

The *flexor digitorum superficialis* of the hindlimb (FDSH) of *R. unicornis* is like that of other perissodactyls. That of *C. simum* is more peculiar by being entirely tendinous, and its tendon merges with that of the *profundus* in the adult specimen. In our neonate *C. simum*, both *flexores digitorum* were entirely fused. The *superficialis* has been noted as being reduced in tapirs, domestic bovids and equids (*Bressou, 1961*; *Barone, 2010*), although *Payne et al. (2005)* noted a relatively high PCSA for that muscle in horses, still not as high as that of the *profundus* (417 vs 666 cm$^2$). *Fisher, Scott & Adrian (2010)* did note that the *superficialis* lacks a distinct muscle belly and present few muscular fibres in *Hippopotamus*, but elephants appear to retain a clear muscular belly (*Miall & Greenwood, 1878*; *Eales, 1928*; *Weissengruber & Forstenpointner, 2004*). Perhaps the *superficialis's* function tends to be transferred to the *profundus* in perissodactyls and artiodactyls due to the larger space for attachment available on the caudal tibia, a tendency that is most extreme in *C. simum*. The origin of the *superficialis* is similar in all the clades we compared, except in elephants where the origin is more superficial, from fascia covering the joint capsule of the knee. We observed in all specimens a complete fusion of the *flexores digitorum lateralis* and *medialis* into a single *flexor digitorum profundus* of the hindlimb (FDPH),consistent with previous observations in rhinos (*Haughton, 1867*; *Beddard & Treves, 1889*). Rhinos seem unique in that regard, as in other perissodactyls, *Hippopotamus*, domestic bovids and elephants, those muscles are separated but share their insertion tendons. The *tibialis caudalis* is absent in rhinos and tapirs and reduced in horses, but is present in *Hippopotamus* and elephants (*Murie, 1871*; *Miall & Greenwood, 1878*; *Beddard & Treves, 1889*; *Eales, 1928*; *Bressou, 1961*; *Barone, 2010*; *Fisher, Scott & Adrian, 2010*).

## Quantitative characterisation

A total of 3,678 measurements were taken, from 270 muscles of four individual rhinoceroses (see Table S1). This includes 2,029 measurements of fascicle length, 909

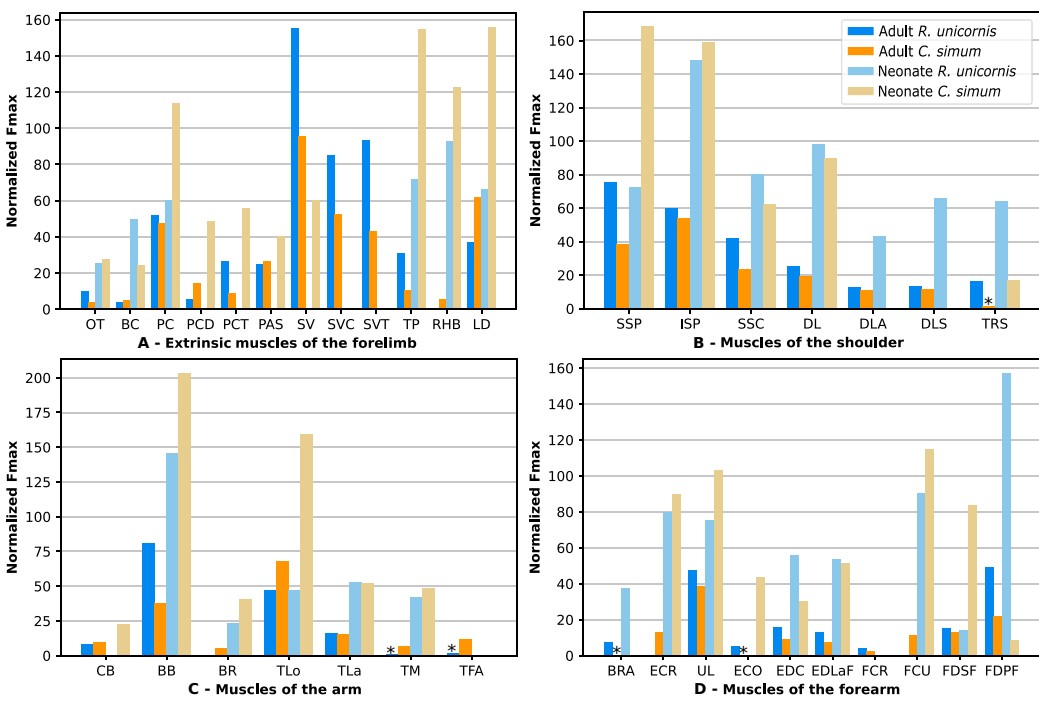

**Figure 10 Normalized Fmax of the muscles of the forelimb of our four rhinoceroses.** Fmax was normalized by dividing it by the total weight of the animal, in Newtons (N). *: Normalized Fmax calculated but close to 0%. Muscle acronyms are in Table 2. Muscle categories follow *Barone (2010)*. (A) Extrinsic muscles of the forelimb. (B) Muscles of the shoulder. (C) Muscles of the arm. (D) Muscles of the forearm.                         

pennation angles, 264 muscle bellies weighed and measured, as well as 102 tendons. In the adult *R. unicornis*, the grand mean of the fascicle lengths of all muscles was 19.19 cm for the forelimb and 14.11 cm for the hindlimb. In the adult *C. simum*, it was 19.03 cm and 22.23 cm for forelimb and hindlimb respectively. In the neonate *R. unicornis*, it was 7.37 cm and 7.54 cm. In the neonate *C. simum*, it was 9.73 cm and 9.07 cm.

### Forelimb

In the adult *C. simum*, the *serrati ventrales* (SVC and SVT) were partially damaged due to the separation of the limb from the body, but a sufficient part was salvaged to calculate average fascicle lengths and pennation angles. The masses of both muscles were extrapolated from their mass in *R. unicornis*, we considered that they take up the same proportion of the animal's mass. In the adult *C. simum*, only the humeral head of the *flexor digitorum superficialis* (FDSF) could be measured, due to damage to the ulnar head during dissection. The strongest muscles in the forelimb of the adult *R. unicornis* were the *serrati ventrales* (SVC and SVT), which were both close to being able to exert a force greater than the body weight of the rhino (85% for the *pars cervicis*, 93% for the *pars thoracis*, Fig. 10, Table S1). The *biceps brachii* (BB), *supraspinatus* (SSP), *infraspinatus* (ISP) and *pectorales* (PC) as a whole each were capable of exerting a force greater than half the body weight. The strongest muscle in *C. simum* was the long head of the triceps (TLo, 68% of body weight, Fig. 10, Table S1). The *latissimus dorsi* (LD), *infraspinatus* (ISP)

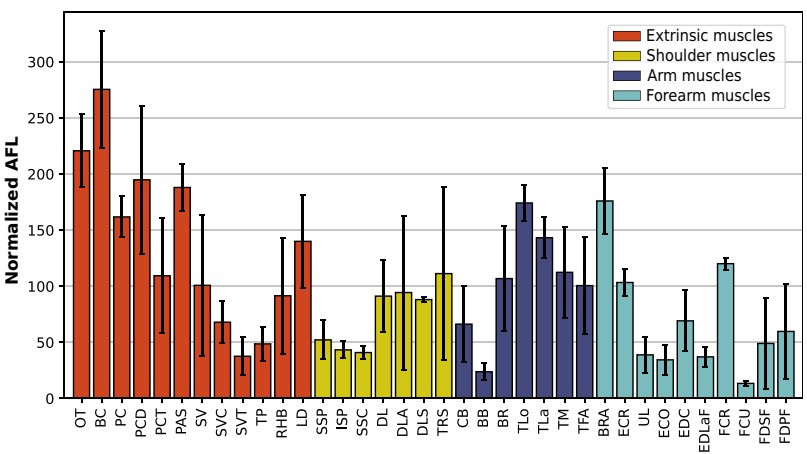

**Figure 11 Normalized average fascicle length (%) of the muscles of the forelimb, averaged from the four specimens for each muscle.** Error bars correspond to one standard deviation above and below the mean. Muscle acronyms are in Table 2.               

and *serratus ventralis pars cervicis* (SVC) were also able to exert a force greater than half the body weight. There was no statistical difference in average normalized Fmax between the adult specimens of the two species for the muscles of the forelimb (Student's t-test: t = 1.20 $p = 0.24$).

In the forelimb of the neonate *R. unicornis* (Fig. 10, Table S1), three muscles were able to exert an estimated maximal force greater than body weight: the *flexor digitorum profundus* (FDPF, 157%), *infraspinatus* (ISP, 148%), and *biceps brachii* (BB, 145%). In the forelimb of the neonate *C. simum* (Fig. 10, Table S1), there were 10 such muscles: the *biceps brachii* (BB, 203%), *supraspinatus* (SSP, 168%), *triceps brachii caput longum* (TLo, 160%), *infraspinatus* (ISP, 160%), *latissimus dorsi* (LD, 156%), *trapezius* (TP, 155%), *rhomboidei* (RHB, 123%), *flexor carpi ulnaris* (FCU, 115%); *pectorales* (PC, 114%) and *ulnaris lateralis* (UL, 103%). There was no statistical difference in average normalized Fmax between the neonate specimens of the two species for the muscles of the forelimb (t = −0.46, $p = 0.65$). Neonate individuals had a greater average normalized Fmax than adults of the same species for the muscles of the forelimb (t = −5.75 for *C. simum*, t = −4.17 for *R. unicornis*, $p < 0.001$ for both species). Almost all muscles indeed presented a greater relative maximal force capacity in neonates, with the exception of the *supraspinatus* (SSP) and *flexor digitorum superficialis* (FDSF) in *R. unicornis* and the *serrati ventrales* (SV) and *flexor digitorum profundus* (FDPF) in *C. simum*.

In the forelimb, the muscles with the relatively longest fascicles were the *omotranversarius* (OT) and *brachiocephalicus* (BC, Fig. 11). Among the extrinsic muscles, the *serrati ventrales* (SV, SVC, SVT) and the *trapezius* (TP) had particularly low normalized AFL. The *infraspinatus* (ISP), *supraspinatus* (SSP) and *subscapularis* (SSC) had a similar normalized AFL, shorter than the other muscles of the shoulder. The *biceps brachii* (BB) showed a relatively low normalized AFL compared to the *triceps* (TLo, TLa, TM), the *tensor fasciae antebrachiae* (TFA) and the *brachialis* (BR). The muscles of the

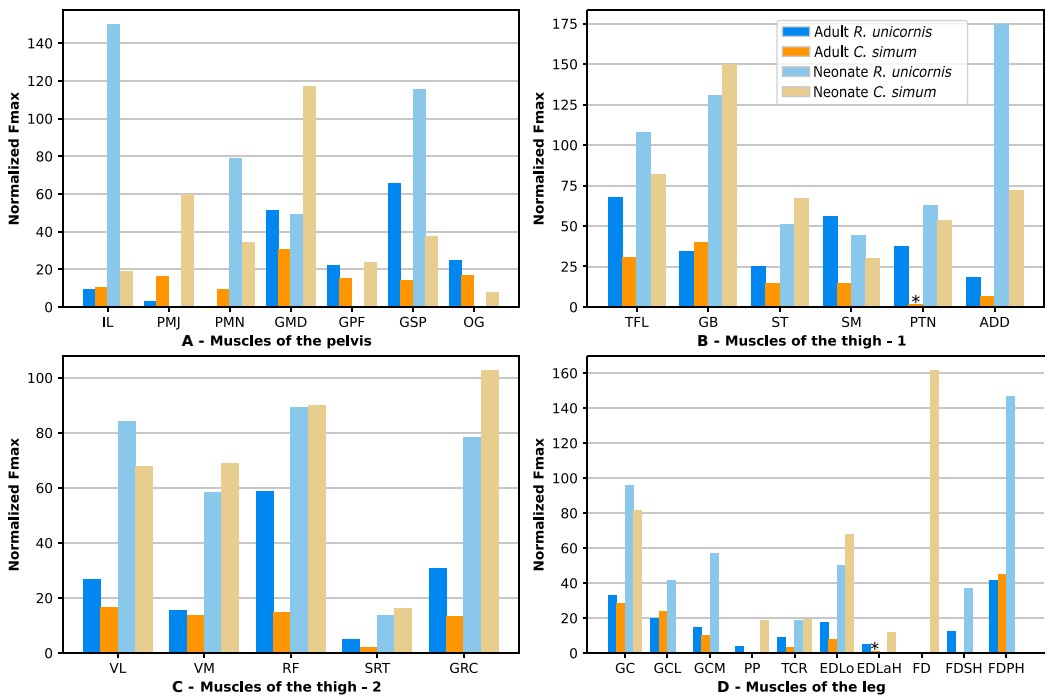

**Figure 12 Normalized Fmax of the muscles of the hindlimb of our four rhinoceroses.** Fmax was normalized by dividing it by the total weight of the animal, in Newtons (N). *: Normalized Fmax calculated but close to 0%. FD: *flexores digitorum*, other muscle acronyms are in Table 3. Muscle categories follow *Barone (2010)*, thigh muscles are divided for readability reasons. (A) Muscles of the pelvis. (B) Muscles of the thigh - *tensor fasciae latae, gluteobiceps, semitendinosus, semimembranosus, pectineus, adductores.* (C) Muscles of the thigh - *quadriceps femoris, sartorius, gracilis.* (D) Muscles of the leg. Value for the *gluteobiceps* (GB) in the adult *R. unicornis* is incomplete.

forearm generally had shorter normalized AFL than average, except for the *brachioradialis* (BRA), the *extensor carpi radialis* (ECR) and the *flexor carpi radialis* (FCR).

### Hindlimb

Due to difficulties in the assignment of the homologies of the *fibulares* between our specimens, their values are not reported. In the hindlimb of the adult *R. unicornis* (Fig. 12, Table S1), no muscle could exert an estimated force greater than body weight. Five could exert a force greater than half of body weight: the *tensor fasciae latae* (TFL, 67%), *gluteus superficialis* (GSP, 65%), the *rectus femoris* (RF, 59%), *semimembranosus* (SM, 56%) and *gluteus medius* (GMD, 51%). In the adult *C. simum* (Fig. 12, Table S1), no muscle could exert a force greater than 50% of body weight; the strongest muscle was the *flexor digitorum profundus* (FDPH, 45%). On average, the muscles of the hindlimb of the adult *R. unicornis* had a greater normalized Fmax than those of the adult *C. simum* (t = 2.33, $p < 0.05$).

Six muscles could exert an estimated force greater than body weight in the neonate *R. unicornis* (Fig. 12, Table S1). Those were the *adductores* (174%), *illiacus* (150%), *flexor digitorum profundus* (FDSH, 146%), *gluteobiceps* (GB, 131%), *gluteus superficialis* (GSP, 116%) and *tensor fasciae latae* (TFL, 108%). In the neonate *C. simum* (Fig. 12, Table S1),

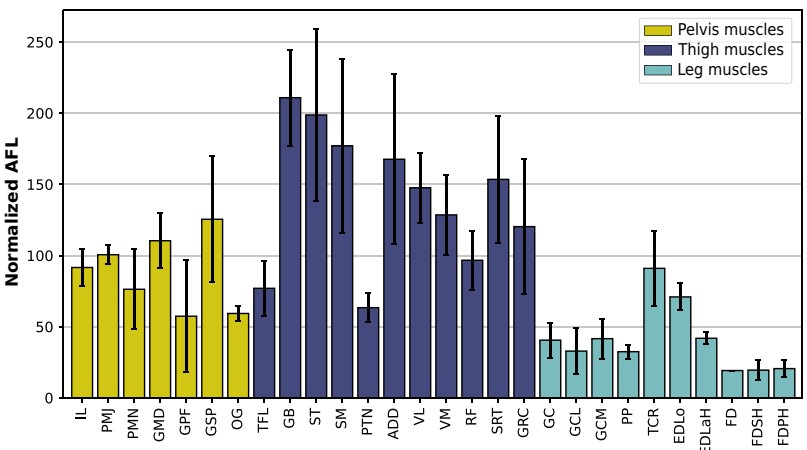

**Figure 13 Normalized average fascicle length (%) of the muscles of the hindlimb, averaged from the four specimens for each muscle.** Error bars correspond to one standard deviation above and below the average. FD: *flexores digitorum*, other muscle acronyms are in Table 3.

the strongest muscles were the *flexores digitorum* (FD, 161%), *gluteobiceps* (GB, 150%), *gluteus medius* (GMD, 117%) and *gracilis* (GRC, 103%). The *flexor digitorum superficialis* and *flexor digitorum profundus* were not yet separated in the neonate *C. simum* and were thus measured as one. There was no statistical difference in average normalized Fmax between the neonate specimens of the two species (t = 0.98, $p = 0.34$). Neonate individuals again had a greater average normalized Fmax than the adults of the corresponding species (t = −5.46 for *C. simum*, t = −4.57 for *R. unicornis*, $p < 0.001$ for both species). This was true of all the individual muscles, except the *gluteus medius* (GMD) and *semimembranosus* (SM) in *R. unicornis*, and the *obturator et gemelli* (OG) in *C. simum*.

In the hindlimb, the muscles with the relatively longest fascicles generally were the muscles of the thigh, except the *pectineus* (PTN) and the *tensor fasciae latae* (TFL, Fig. 13). The *gluteus superficialis* (GSP) and the *gluteus medialis* (GMD) had a normalized AFL longer than the *gluteus profundus* (GPF). The muscles of the leg all had a particularly short normalized AFL, except for the *tibialis cranialis (TCR)* and the *extensor digitorum longus* (EDLo).

## DISCUSSION

In the first section of the Discussion, we discuss the morphofunctional importance of the differences of qualitative myology observed between the various clades in the previous section, and draw conclusions on the relevance of the concept of graviportality from a muscular point of view. The second section is devoted to quantitative architecture and potential adaptations to sustain and move an important body mass, comparing with quantitative data for horses and tapirs. Additional quantitative comparisons were made with the muscle mass reported in *R. unicornis* by *Haughton (1867)*, in Supplemental Data (Table S2, File S2). The third section presents the ontogenetic trends that may be present in our sample.

## General morphological considerations

From a qualitative point of view, we found few differences between the limb myology of *R. unicornis* and *C. simum*. No differences were found that could be obviously linked to their differences in general morphology or habitat. It seems that their large adult body mass drives most of the adaptations found in rhinoceroses' myology. Rhinos present a very similar organization of the limb muscles to the other perissodactyls, pointing at many characteristics inherited from a common ancestor (e.g. the *omotransversarius* inserting at least partly on the humerus, a single-headed *subscapularis*, the absence or great reduction of the *tibialis caudalis* and of the *soleus*). Many of the traits observed in rhinos can be linked to their high degree of cursoriality for their size. The *omotransversarius* and *brachiocephalicus* present similar paths and myology, being non-pennate with very long fascicles (Fig. 11, Table S1). This could increase the speed and working range of contraction, and permit efficient protraction of the forelimb during swing phase. The more distal insertion of the *omotransversarius* compared to that of horses and tapirs, on the humerus, would allow it to act with a greater lever arm on the whole limb, which may be useful to protract a relatively heavy forelimb at the cost of a slower speed of rotation. The *illiacus* and *psoas major* are the main muscles involved in protraction of the hindlimb, and present a similar organization to their forelimb counterparts, with long fascicles but a relatively low PCSA, as they only act on the limb and not on the whole animal beyond the pelvis. *Mallet et al. (2019)* noted that the lesser trochanter is more distal in rhinoceroses than in horses, giving the *illiacus* and *psoas major* a greater lever arm for limb protraction, similar to the humeral insertion of the *omotransversarius*; thus the protractors of both limbs present similar adaptations in terms of architecture and insertion. This is not found in hippopotamuses and elephants, indicating that this is likely an adaptation to both heavy weight and high speed. Overall, the propulsor muscles of the hindlimb, especially the gluteal muscles, the *gluteobiceps* and the *semitendinosus* present many similarities with other perissodactyls. This is likely an organisation retained from a cursorial common ancestor, meaning that propulsion of the body by the hindlimb is likely conducted in a similar way in all perissodactyls. Compared to other perissodactyls, the more distal insertion of the *subclavius* in the forelimb, and of the *adductores* in the hindlimb, likely give those muscles a stronger lever arm to adduct their respective limb.

In contrast, contrary to our hypothesis that rhinos would share significant traits with elephants and hippopotamuses as well, very few convergences were identified between the three clades defined as graviportal. The main one is the unique insertion of the *supraspinatus* muscle, on the greater tubercle of the humerus, also shared with suids and giraffes. The latter also reach a large adult body mass (up to 1,930 kg for males, *Skinner & Mitchell, 2011*). Because the *supraspinatus* is one of the most important extensors of the shoulder, perhaps a unique, stronger insertion on the humerus concentrates the force generated on an efficient point for shoulder extension, allowing for a greater extension capacity in heavy species. A double insertion on both tuberosities, on the other hand, could allow more shoulder stability for lighter, more cursorial species such as horses and bovids. Another characteristic common to heavy mammals seems to be a strong *tensor fasciae*

*latae*. This could provide a forceful extension of the knee, although in the absence of true quantitative characterisation of *Hippopotamus* and elephant hindlimb muscles, this remains hypothetical. Our results indicate that from a myological point of view, rhinos, elephants and hippos can hardly be classified together as graviportal. This is especially true considering that rhinos do not show the more columnar limbs and the absence of a galloping gait generally thought to be characteristic of graviportality (*Gregory, 1912*; *Alexander & Pond, 1992*; *Mallet et al., 2019*). It seems that the mammalian musculoskeletal system adapted to heavy body weight with fundamentally different paths, the set of adaptations displayed varying depending on the phylogenetic history of the clade and on the other constraints on which it may be submitted (e.g. cursoriality for rhinoceroses or semi-aquatic lifestyle for *Hippopotamus*).

Most muscles involved in distal limb supination and pronation are absent or greatly reduced in rhinos. This is similar to what is generally observed in ungulates, as active muscle-driven pronation and supination are more restricted than in carnivores, primates, or in placental mammals ancestrally (*Iwaniuk & Whishaw, 2000*; *Anderson & Werdelin, 2003*; *Polly, 2008*; *Bonnan et al., 2016*). Indeed, ungulate forelimbs are almost exclusively used for locomotion, and thus are expected to be specialized in that way. Other mammals may use their forelimbs for various tasks (e.g. prey capture, grasping) that require a greater range of pronation and supination.

Several myological traits seem to present significant intraspecific variability in the species we studied (e.g. insertion of the gluteus superficialis on the fibula, presence of a vastus intermedius). Muscular architecture is also likely to vary greatly from one specimen of the same species to another. Our sample size sadly prevents us from further addressing the question of intraspecific variability. Rhinoceroses being rare and heavy animals, the preservation and transport of their body alone is a challenge making it extremely difficult to get specimens to dissect. Likewise, all specimens were captive-bred, which may have led to alterations of their muscles.

## Architectural adaptations to weight-bearing
### Forelimb

In rhinos, the strongest muscles are clearly the more proximal ones in the limb (Table 4). In adults, the total PCSA of the muscles of the forearm is approximately 45% of that of the extrinsic muscles, whereas it is 85% in horses (Table 4). Most of the muscles used by rhinos to sustain their large body mass are therefore located in the proximal region. This has a double advantage: first, it allows the muscles to grow larger in volume due to the greater space available in the proximal limb and the trunk. Second, it concentrates muscular mass in the proximal segments of the limb, avoiding having heavier distal segments which, by lever effect, would be harder to move than the proximal segments for a given mass (*Alexander, 1977*; *Payne et al., 2005*; *Smith et al., 2006*, *2007*).

The *serrati ventrales* are particularly strong in rhinoceroses, reflecting the fact that they are the main muscles supporting the thorax between the limbs. An interesting difference between horses and rhinos is the relative PCSA of the *serrati ventrales thoracis* and *cervicis*. The latter is eight times as forceful as the former in horses, suggesting that

**Table 4 Comparison of the PCSA (in cm$^2$) between our specimens and specimens of *Equus caballus* and *Tapirus indicus*, for the muscles of the forelimb.**

| | Muscle | E. caballus adult PCSA | % | T. indicus juvenile PCSA | % | CS adult PCSA | % | RU adult PCSA | % | CS neonate PCSA | % | RU neonate PCSA | % |
|---|---|---|---|---|---|---|---|---|---|---|---|---|---|
| EXTRINSIC | PC | 160.0 | 123.6 | ND. | | 335.3 | 153.0 | 350.0 | 112.5 | 17.4 | 106.8 | 8.5 | 82.4 |
| | PCD + PCT | 77.0 | 59.5 | ND. | | 161.8 | 73.8 | 218.6 | 70.3 | 16.1 | 98.6 | ND. | |
| | PCA + SU | 83.0 | 64.1 | ND. | | 185.7 | 84.7 | 166.5 | 53.5 | 6.2 | 38.2 | ND. | |
| | SVC | 72.0 | 55.6 | ND. | | 372.3 | 169.8 | 575.7 | 185.1 | ND. | | ND. | |
| | SVT | 577.0 | 445.8 | ND. | | 303.3 | 138.4 | 629.3 | 202.3 | ND. | | ND. | |
| | BC-OT | 62.0 | 47.9 | ND. | | 61.8 | 28.2 | 91.0 | 29.2 | 8.0 | 48.7 | 10.5 | 102.4 |
| | TP | 42.0 | 32.4 | ND. | | 75.4 | 34.4 | 208.8 | 67.1 | 23.8 | 145.7 | 10.1 | 98.0 |
| | LD | 53.0 | 40.9 | ND. | | 437.8 | 199.8 | 248.6 | 79.9 | 24.0 | 146.7 | 9.3 | 90.6 |
| | RHB | 39.0 | 30.1 | ND. | | 39.1 | 17.8 | ND. | | 18.8 | 115.4 | 13.0 | 126.5 |
| | EXT. AV. | 129.4 | | | | 219.2 | | 311.1 | | 16.3 | | 10.3 | |
| SHOULDER | TRS | ND. | | 7.4 | 23.7 | 11.7 | 6.1 | 110.7 | 37.3 | 2.6 | 17.1 | 9.0 | 69.5 |
| | DL | ND. | | 10.0 | 32.0 | 137.1 | 71.0 | 169.8 | 57.2 | 13.8 | 90.6 | 13.8 | 105.8 |
| | SSC | ND. | | 41.3 | 132.3 | 165.0 | 85.4 | 284.8 | 96.0 | 9.6 | 62.9 | 11.3 | 86.5 |
| | ISP | ND. | | 52.1 | 166.9 | 380.7 | 197.1 | 406.8 | 137.1 | 24.5 | 160.0 | 20.8 | 159.9 |
| | SSP | 150.3 | | 45.3 | 145.1 | 271.1 | 140.4 | 511.0 | 172.3 | 25.9 | 169.4 | 10.2 | 78.3 |
| | SH. AV. | | | 31.2 | | 193.1 | | 296.6 | | 15.3 | | 13.0 | |
| ARM | BB | 244.8 | 211.1 | 24.1 | 120.7 | 268.6 | 159.4 | 544.8 | 262.7 | 31.2 | 231.7 | 20.5 | 234.1 |
| | CB | ND. | | 4.9 | 24.5 | 66.8 | 39.7 | 55.2 | 26.6 | 3.5 | 25.7 | ND. | |
| | BR | ND. | | 10.8 | 54.1 | 36.3 | 21.6 | ND. | | 6.2 | 46.2 | 3.3 | 37.4 |
| | TLo | 168.3 | 145.1 | 58.8 | 294.5 | 478.9 | 284.2 | 319.7 | 154.1 | 24.5 | 182.0 | 6.7 | 76.5 |
| | TLa | 38.4 | 33.1 | 16.1 | 80.6 | 111.8 | 66.4 | 111.5 | 53.8 | 8.0 | 59.4 | 7.4 | 84.5 |
| | TM | 12.3 | 10.6 | 5.1 | 25.5 | 48.5 | 28.8 | 5.8 | 2.8 | 7.4 | 55.0 | 5.9 | 67.4 |
| | ARM. AV. | 116.0 | | 20.0 | | 168.5 | | 207.4 | | 13.5 | | 8.8 | |
| FOREARM | BRA | ND. | | 1.0 | 7.7 | 2.9 | 3.2 | 51.8 | 35.0 | ND. | | 5.3 | 46.6 |
| | ECO | 19.1 | 17.4 | 7.3 | 56.0 | 2.0 | 2.2 | 35.0 | 23.6 | 7.0 | 60.3 | ND. | |
| | EDC | 36.3 | 33.1 | 5.7 | 43.7 | 63.3 | 68.4 | 105.9 | 71.5 | 4.7 | 40.5 | 7.9 | 69.6 |
| | EDL | 12.1 | 11.0 | 4.6 | 35.3 | 53.1 | 57.4 | 88.0 | 59.4 | 7.9 | 68.1 | 7.6 | 66.9 |
| | ECR | 99.3 | 90.7 | 9.6 | 73.6 | 91.5 | 98.9 | ND. | | 13.8 | 119.0 | 11.3 | 99.5 |
| | FCU | 133.9 | 122.2 | 10.6 | 81.3 | 82.0 | 88.6 | ND. | | 17.7 | 152.6 | 12.7 | 111.8 |
| | FCR | 18.5 | 16.9 | 9.5 | 72.8 | 19.0 | 20.5 | 27.0 | 18.2 | ND. | | ND. | |
| | UL | 193.8 | 176.9 | 24.7 | 189.4 | 273.0 | 295.1 | 322.3 | 217.7 | 15.9 | 137.1 | 10.6 | 93.3 |
| | FD | 363.3 | 331.7 | 44.4 | 340.4 | 245.8 | 265.7 | 406.3 | 274.4 | 14.2 | 122.4 | 24.1 | 212.2 |
| | FA. AV. | 109.5 | | 13.0 | | 92.5 | | 148.0 | | 11.6 | | 11.4 | |
| | Grand total | 2,655.4 cm$^2$ | | 393.3 cm$^2$ | | 4,781.8 cm$^2$ | | 6,045 cm$^2$ | | 352.8 cm$^2$ | | 239.8 cm$^2$ | |
| | | Without shoulder | | Without extrinsic | | | | | | | | | |

**Note:**
Data for horses were all collected on adult specimens, and come from *Payne, Veenman & Wilson (2005)* for the extrinsic muscles ($n = 7$), from *Watson & Wilson (2007)* for the *triceps, biceps* and *supraspinatus* ($n = 2$) and from *Brown et al. (2003)* for the muscles of the forearm ($n = 7$). Tapir data are from *MacLaren & McHorse (2020)*, and were gathered on one juvenile individual. CS: *Ceratotherium simum*, RU: *Rhinoceros unicornis*, AV.: average, EXT.: extrinsic muscles, SH.: muscles of the shoulder, ARM.: Muscles of the arm, FA.: muscles of the forearm, ND.: no data. Data were normalized ("%" column) by dividing the PCSA by the average of the muscle group and multiplying by 100.

horses have a need for an important *serrati ventrales thoracis* to support their thorax, but do not need an equally important *serrati ventrales cervicis* to support their heads. In rhinos, the two muscles have an equivalent PCSA. This is likely because rhinos have a more massive head than horses, necessitating a proportionally stronger *serrati ventrales cervicis* to sustain it. Additionally, rhinos, especially *C. simum*, carry their head very low with regard to the axis of the vertebral column, contrary to horses. Horses may therefore use their *rhomboideus cervicis* more than their *serrati ventrales cervicis* for supporting their head. The *rhomboideus* is indeed proportionally weaker in our adult *C. simum* than in horses. We sadly could not measure the *rhomboideus* in *R. unicornis*. The average fascicle length and pennation angle of both *serrati ventrales* is similar in rhinos (Figs. 11; Table S1), whereas in horses the *cervicis* has ten times longer fascicles than the *thoracis*. *Payne, Veenman & Wilson (2005)* noted a particular architecture of the *serrati ventrales thoracis* in horses, with a 45° angle of pennation and 4.9 cm-long fascicles. It is remarkable that we found very similar values in our adult *R. unicornis* (44°, 4 cm), with *C. simum* presenting even shorter fascicles (31°, 1 cm). They hypothesized that this architecture improves resistance to gravity, by increasing muscle force output at the expense of range of motion. Our results are consistent with this hypothesis: the *serrati ventrales thoracis* seems to be specialized in supporting the massive trunk of rhinoceroses, and its action in protraction of the limb seems greatly reduced, but passed on to the effective pair of the synergistic *omotransversarius* and *brachiocephalicus*. The *serratus ventralis cervicis* seems specialized in a similar way to support the heavy head. The *latissimus dorsi* is strong compared with that of horses. When the forelimb is in stance phase, its main function is to support and decelerate the body; its greater PCSA is likely necessary given the greater body mass of rhinos.

The *infraspinatus* and *supraspinatus* are the strongest muscles in the shoulder region, reflecting their important actions in extension and stabilization of this articulation. Those muscles, as well as the *subscapularis*, present noticeably short fascicles, suggesting that they are specialized in generating a strong force but only producing a small displacement of the joint. Their action is most likely to lock the shoulder joint firmly into place (i.e. acting as stabilizers; or resisting flexion under gravity). The *biceps brachii* is also a strong muscle with short fascicles, which is likely due to its action in shoulder flexion, rather than its action as a flexor of the forearm. The *biceps* may also be important in the protraction of the limb during the initiation of the swing phase, as in horses where it stores elastic energy during the stance phase that it can then recover with less metabolic cost for the animal (*Watson & Wilson, 2007*). This is consistent with the prior observation that the insertion area of the *biceps brachii* on the radius is more robust in the heaviest species of rhinos (*Mallet et al., 2019*). The *triceps brachii*, especially its *caput longum*, is also among the strongest muscles, and benefits from a long olecranon in rhinoceroses, creating a large lever arm (*Maynard Smith & Savage, 1956*; *Mallet et al., 2019*). Its fascicles are longer than those of the *biceps* and the extensors and stabilizers of the shoulder, likely related to the length of the olecranon, balancing length change costs and benefits from fascicle lengths and lever arms (*Gans & De Vree, 1987*). The *triceps brachii*'s combined actions with the *biceps*, the *infraspinatus*, and the *supraspinatus* are probably of great

importance to support the limb against gravity. Of similar actions are the *pectorales*, as their large maximal force output should help maintain the limb in adduction; the more distal insertion of the *subclavius*, on the humerus rather than the scapula, may provide this muscle with a greater lever arm in this regard. *Mallet et al. (2019)* noted a substantial development of the lesser tubercle in heavy rhinos (including our two species), and inferred from horses that this was due to the medial insertion of the *supraspinatus*. That insertion is absent in rhinoceroses; the distinct development of this region may instead be linked to the considerable forces imposed by the combined *pectoralis ascendens* and *subclavius*.

The pattern observed in the muscles of the forearm is similar to that of horses and tapirs. The *flexores digitorum* are the strongest muscles, generally followed by the *ulnaris lateralis* and the *flexor carpi ulnaris*. In horses, all of those muscles act in synergy to initiate the stance phase and decelerate the body; it is likely that their role is the same in rhinos (*Harrison et al., 2010*). The *extensor digitorum communis* and *lateralis* and the *extensor carpi radialis* are stronger in rhinos than in tapirs and horses. These muscles are involved in the stability of all the articulations of the manus; it is therefore logical that they have to be proportionally stronger in heavier animals. The tendons of all the muscles inserting on the digits are generally of similar length and apparent robustness for all three digits, which is concordant with the tridactyly of rhinoceroses and that forces are evenly distributed between the toes (*Panagiotopoulou, Pataky & Hutchinson, 2019*).

This general specialization of the forelimb for body weight support is consistent with what is generally known in quadrupedal mammals and especially ungulates, and is here taken to another extreme by the heavy weight of rhinoceroses. The muscles of the forelimb had a total PCSA higher than those of the hindlimb in all our specimens, whereas in highly cursorial horses, the hindlimb has a higher total PCSA than the forelimb, although PCSA data are absent for four muscles of the horse forelimb (Tables 4, 5). All of these inferences are consistent with the higher degree of integration linked to mass observed between the bones of the forelimb in rhinoceroses, compared to those of the hindlimb (*Mallet et al., 2020*). The large PCSA shown by the muscles of the forelimb, required for body support, may drive the bones' shape towards similar adaptations (e.g. larger insertion areas) and thus increase the degree of integration between them.

### Hindlimb

The average PCSA of the muscles remained roughly constant in the different segments of the hindlimb (Table 5). This is in stark contrast with *E. caballus*, where the muscles of the leg confer greater forces, on average, than the muscles of the pelvis, which is consistent with the pattern observed in the forelimb. This considerable force-generating capacity of the equine distal hindlimb is driven by the *flexores digitorum*, which have a combined PCSA of 1120 cm² , which is much stronger than what is observed for any other muscles in horses or rhinoceroses. Overall, our adult *R. unicornis* had a total PCSA in the hindlimb equivalent to that of horses, and *C. simum*'s PCSA was 60% of that of horses, despite horses being four times lighter than both our specimens. This is most likely due to the high degree of cursorial specialization observed in horses, further exacerbated by

Table 5 Comparison of the PCSA values (in cm²) between our specimens and specimens of *Equus caballus*, for the muscles of the hindlimb.

| | Muscle | Equus adult | | CS adult | | RU adult | | CS neonate | | RU neonate | |
|---|---|---|---|---|---|---|---|---|---|---|---|
| | | PCSA | % | PCSA | % | PCSA | % | PCSA | % | PCSA | % |
| **PELVIS** | **GSP** | 60.0 | 48.8 | 100.0 | 88.0 | 441.3 | 223.4 | 5.7 | 87.5 | 16.3 | 117.7 |
| | **GMD** | 398.0 | 324.0 | 216.2 | 190.2 | 346.9 | 175.6 | 18.0 | 274.7 | 6.9 | 49.8 |
| | **GPF** | 108.0 | 87.9 | 107.6 | 94.6 | 147.7 | 74.8 | 3.6 | 54.8 | *ND.* | |
| | **PMJ** | 56.0 | 45.6 | 115.1 | 101.3 | 19.9 | 10.1 | 9.2 | 140.5 | *ND.* | |
| | **PMN** | 61.0 | 49.7 | 65.5 | 57.6 | *ND.* | | 5.3 | 80.5 | 11.0 | 80.0 |
| | **IL** | 54.0 | 44.0 | 73.4 | 64.6 | 63.0 | 31.9 | 2.9 | 44.1 | 21.1 | 152.6 |
| | **OG** | *ND.* | | 117.9 | 103.7 | 166.2 | 84.1 | 1.2 | 17.8 | *ND.* | |
| | **PLV. AV.** | 122.8 | | 113.7 | | 197.5 | | 6.5 | | 13.8 | |
| **THIGH** | **TFL** | 140.0 | 85.3 | 213.8 | 198.5 | 455.4 | 201.5 | 12.6 | 112.6 | 15.2 | 132.7 |
| | **GB** | 294.0 | 179.1 | 283.0 | 262.8 | 232.5 | 102.9 | 23.1 | 206.4 | 18.3 | 160.6 |
| | **ST** | 144.0 | 87.7 | 101.2 | 93.9 | 166.8 | 73.8 | 10.3 | 92.0 | 7.2 | 63.1 |
| | **SM** | 106.0 | 64.6 | 101.0 | 93.8 | 378.0 | 167.3 | 4.6 | 41.3 | 6.2 | 54.1 |
| | **VL** | 105.0 | 64.0 | 117.3 | 109.0 | 179.5 | 79.4 | 10.4 | 93.1 | 11.8 | 103.5 |
| | **VI** | 45.0 | 27.4 | *ND.* | | *ND.* | | *ND.* | | *ND.* | |
| | **VM** | 148.0 | 90.2 | 95.3 | 88.5 | 105.0 | 46.5 | 10.6 | 94.9 | 8.2 | 71.8 |
| | **RF** | 552.0 | 336.2 | 104.9 | 97.4 | 396.0 | 175.2 | 13.8 | 123.6 | 12.5 | 109.6 |
| | **PTN** | 78.0 | 47.5 | 11.2 | 10.4 | 211.0 | 93.4 | 8.2 | 73.6 | 8.8 | 77.2 |
| | **SRT** | 12.0 | 7.3 | 15.0 | 13.9 | 33.4 | 14.8 | 2.5 | 22.2 | 1.9 | 16.6 |
| | **GRC** | 135.0 | 82.2 | 93.7 | 87.1 | 206.4 | 91.3 | 15.8 | 141.2 | 11.0 | 96.2 |
| | **ADD** | 211.0 | 128.5 | 48.0 | 44.5 | 121.7 | 53.9 | 11.1 | 99.0 | 24.5 | 214.6 |
| | **TH. AV.** | 164.2 | | 107.7 | | 226.0 | | 11.2 | | 11.4 | |
| **LEG** | **GC** | 298.0 | 109.0 | 200.6 | 165.1 | 222.2 | 162.6 | 12.5 | 135.6 | 13.4 | 110.0 |
| | **PP** | 70.0 | 25.6 | *ND.* | | 26.9 | 19.7 | 2.9 | 31.4 | *ND.* | |
| | **TCR** | 73.0 | 26.7 | 24.2 | 19.9 | 58.4 | 42.7 | 2.9 | 31.7 | 2.6 | 21.1 |
| | **EDLo** | 54.0 | 19.7 | 56.6 | 46.6 | 117.1 | 85.7 | 10.4 | 112.9 | 7.1 | 57.8 |
| | **EDLaH** | 26.0 | 9.5 | 8.3 | 6.8 | 31.3 | 22.9 | 1.8 | 19.4 | *ND.* | |
| | **FD** | 1,120.0 | 409.5 | 317.8 | 261.6 | 364.1 | 266.4 | 24.8 | 269.0 | 25.8 | 211.0 |
| | **LEG AV.** | 273.5 | | 121.5 | | 136.7 | | 9.2 | | 12.2 | |
| | **Grand total** | 4,348.0 cm² | | 2,587.5 cm² | | 4,490.8 cm² | | 224.0 cm² | | 229.7 cm² | |

**Note:**
Data for horses were all collected on adult specimens, and come from *Payne et al. (2005)* (*n* = 7). CS: *Ceratotherium simum*, RU: *Rhinoceros unicornis*, AV.: average, PLV.: Muscles of the pelvis, TH.: muscles of the thigh, ND.: no data. Data were normalized ("%" column) by dividing the PCSA by the average of the muscle group and multiplying by 100.

domestication. Most of the horses dissected in *Payne et al. (2005)* are indeed from breeds used for horse racing, capable of reaching up to 19 m s⁻¹ with a rider on (*Spence et al., 2012*) whereas *C. simum* might reach ~7.5 m s⁻¹ (*Alexander & Pond, 1992*); no empirical data are available for *R. unicornis*. Additionally, our individual of *C. simum* had a generalized weakness at the end of its life, which may have lowered its muscular mass and thus PCSA. This may also explain why it had a lower normalized Fmax than our adult *R. unicornis* in the hindlimb. The forelimb might not have been affected because its weight-bearing role is likely more obligatorily required for a captive animal than the

propulsor role of the hindlimb, which may have prevented muscle atrophy, but this is speculative.

The strongest muscles in the hindlimb are those involved in gravitational support and propulsion of the body, i.e. the gluteal muscles, the *gluteobiceps, semimembranosus, semitendinosus, quadriceps femoris*, as well as the *gastrocnemius* and *flexores digitorum*. An interesting difference from the horse is the greater PCSA of the *gluteus superficialis*, which is even larger than that of the *medius* in both our *R. unicornis*. When the hip is already partially in extension due to the action of the hamstring muscles and of the *gluteus medius*, the *gluteus superficialis* could act as an additional extensor of the limb, and benefit from a longer lever arm than the *gluteus medius*, incurred by the more distal position of the third trochanter compared to the greater trochanter. *Mallet et al. (2019)* reported that in *R. unicornis*, those two trochanters are sometimes linked by a bony bridge, although this was not the case in our specimens. There could therefore be a continuity in the insertion of all the gluteal muscles, and the *superficialis* could act as an extensor after the more proximal *medius* and *profundus* have already partially extended the hip, perhaps explaining why its normalized Fmax is greater in our *R. unicornis* specimens. This shift of action of the *gluteus medius* towards that of the *gluteus superficialis* would explain the reduction in the proximal development of the greater trochanter in heavy rhinos noted by *Mallet et al. (2019)*.

As in horses, the *gluteobiceps, semitendinosus* and *semimembranosus* of our rhinos were all strong muscles, and yet retained relatively long fascicles. This likely reflects a tradeoff between being able to produce a large amount of force and being able to contract rapidly and over a longer distance (*Payne et al., 2005*). Those muscles would therefore be capable of producing a large amount of work useful for body propulsion at a relatively fast speed. This is also the case for the different heads of the *quadriceps femoris*, although their fascicles are slightly shorter, indicating a less extreme potential range and speed of motion at the knee than at the hip. The *tensor fasciae latae* has shorter fascicles and is therefore likely to serve as an antigravity muscle keeping the knee in extension.

The strong *gastrocnemius* and *flexores digitorum profundus* are highly pennate, with long tendons able to store elastic strain energy, an architecture that is not observed in elephants (*Weissengruber & Forstenpointner, 2004*), which do not gallop or trot. This is consistent with the observation that the tuber calcanei remains relatively elongated in rhinos but is shortened in elephants (*Etienne et al., 2020*). In horses, the *flexores digitorum* are four times as strong as the *gastrocnemius*, whereas in both our adult rhinos, the *flexores digitorum* are only 1.6 as strong as the *gastrocnemius*. This may be because the gastrocnemius inserts on the tuber calcanei, a large lever arm. It is thus more capable of acting against gravity than the *flexores digitorum*, perhaps avoiding hyperextension of the ankle, useful for heavy animals like rhinos.

Despite those exceptions most likely linked to the large body mass of rhinos, the pattern observed in the hindlimb in terms of relative PCSA and fascicle length is similar to that of horses (*Payne et al., 2005*, *Crook et al., 2008*). This is consistent with the expectation that the hindlimbs perform a major function in body propulsion, as well as a lesser role in support relative to the forelimbs. Comparisons with quantitative anatomical and

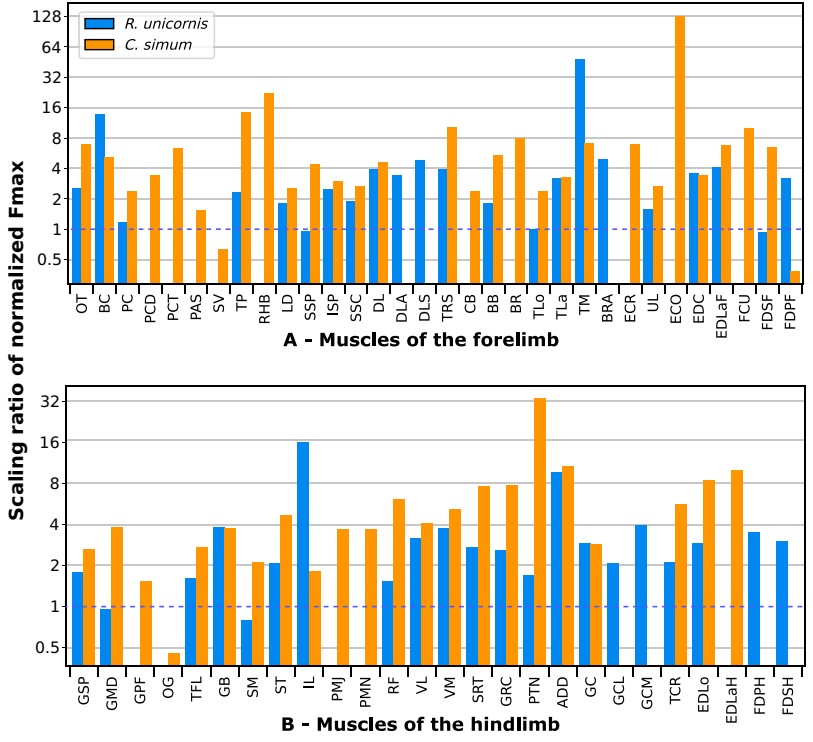

**Figure 14 Ratios of normalized Fmax of the neonate divided by the normalized Fmax of the adult, for both species.** (A) Muscles of the forelimb. (B) Muscles of the hindlimb. The dashed line indicates approximate isometric scaling with body weight (i.e. ratio of 1). Muscles acronyms are in Tables 2 (forelimb) and 3 (hindlimb).

functional data for elephants and hippopotamuses would be interesting to determine if these animals that do not gallop present a different pattern.

## Ontogeny

Our adult specimens were approximately 40 times heavier than our neonates. Several ontogenetic trends could be observed in our sample, although limitations of sample size in this study prevent us from doing a true scaling analysis to quantify how muscles grow in rhinoceroses; a cross-sectional population-level study would be necessary for this. The relative maximal isometric force (Fmax) of almost all muscles suggested a negatively allometric scaling relationship (Fig. 14); i.e. the neonates were able to exert a much greater normalized Fmax than the adults. This is consistent with our initial hypothesis: in general, smaller mammals are expected to have greater Fmax for their size, especially for muscles involved in locomotion (*Carrier, 1995, 1996*; *Herrel & Gibb, 2006*). Weight is expected to scale with linear dimensions cubed whereas PCSA, as an area, scales with linear dimensions squared (*Hildebrand, 1982*; *Hildebrand et al., 1985*; *Biewener, 1989*) and thus strength : weight ratios inevitably decline in large animals via ontogeny or phylogeny. On average, normalized Fmax is 4.38 times greater in the neonate *R. unicornis* than in the adult, and 8.16 times greater in *C. simum*. Again, this difference could be due to the general weakness our adult *C. simum* suffered at the end of its life, or to differences in the term of the pregnancy of the neonates that could affect muscle development.

A few muscles were an exception to the negatively allometric scaling we inferred: the *supraspinatus*, *flexor digitorum superficialis* of the forelimb, *gluteus medius* and *semimembranosus* in *R. unicornis*, and the *serrati ventrales*, *flexor digitorum profundus* of the forelimb and *obturator et gemelli* in *C. simum*. Except for the *obturator et gemelli*, they were all strong muscles involved in either body support or fore/aft motion. This indicates that those muscles probably develop their large Fmax during the growth of the animal and had not yet had the opportunity to do so in very young individuals. Conversely, muscles that have extremely high normalized Fmax in the neonates compared to the adults may start with a relatively high Fmax due to phylogenetic or developmental constraints and then undergo a reduction of muscle volume due to being underused. This is likely the case for the *extensor carpi obliquus* and the *triceps longus caput mediale*.

## CONCLUSIONS

Our study has clarified the appendicular musculature of a clade that was in dire need of a reassessment, and provides the first detailed quantification of muscular architecture for such giant animals. Overall, from a qualitative point of view and contrary to our hypothesis, rhinoceroses' limb musculature presents only a few characteristics linking them with elephants and hippopotamuses, and is instead similar to that of the other perissodactyls, as phylogenetic relationships would predict. In accordance with our hypothesis, rhinos present similar adaptations to running as equids and tapirs do, although with adjustments that probably compensate for their greater body mass, such as more distal insertions for the protractor and adductor muscles. In terms of quantitative architecture, adaptations to heavy weight include stronger forelimb than hindlimb muscles, reflecting the greater emphasis on weight-bearing in the forelimbs of most mammalian quadrupeds. As in most tetrapods, to varying degrees, muscle mass and therefore maximal isometric force are concentrated in the proximal part of both limbs, thus decreasing the mass of the distal segments. Some extensor muscles, mainly in the forelimb (e.g. *serrati ventrales*, *supraspinatus*, *infraspinatus*, *biceps brachii*) display remarkably short fibers and high degrees of pennation that help them to generate strong forces, useful for gravitational support and joint stabilization. Other muscles present longer fascicles and thus a greater speed and working range, but still possess a greater estimated maximal isometric force due to their large volume. Those are mainly propulsor muscles of the hindlimb (e.g. gluteal muscles, *gluteobiceps*, *quadriceps femoris*). Ontogenetic scaling of maximal isometric force is evident in our individuals, with neonates exhibiting a much higher normalized Fmax than adults in almost every muscle. Some extensor muscles are an exception, which indicates that they likely develop their great strength during the growth of the animal. Our results indicates that rhinos, hippos and elephants can hardly be classified together as 'graviportal' from a muscular point of view. It rather seems that rhinos have evolved several traits, in terms of musculoskeletal adaptations (e.g., more distal insertion of protractor and adductor muscles, relatively stronger forelimb for body support and braking during locomotion), to adapt to supporting and moving a body mass of up to several tons without compromising their ability to gallop and achieve somewhat fast speeds, and that these traits could not be

regrouped together under the concept of graviportality. Further studies on elephants and hippopotamuses would prove especially useful to provide an even more comprehensive view of how land vertebrates adapt to sustain a heavy weight, as well as precise biomechanical modelling of the musculoskeletal systems of heavy taxa.

## ACKNOWLEDGEMENTS

We wish to acknowledge very warmly Suzannah Williams and Hinnah Rehman, undergraduates at the Royal Veterinary College who helped greatly with the dissections of the White and Indian rhinos, respectively, and with data collection. We thank Elizabeth Ferrer, Sharon Warner and all those that helped in the Structure and Motion Laboratory (RVC) for assistance in data collection during this project, along with post-mortem room support from Richard Prior. ZSL Whipsnade Zoo, Woburn Safari Park, Munich Hellabrunn Zoo and their staff are wholeheartedly thanked for generously donating the specimens used. Finally, we warmly thank Korakot Nganvongpanit (Chiang Mai University, Thailand) for editorial work, as well as Jamie MacLaren (University of Liège, Belgium) and two anonymous reviewers for their helpful comments and constructive critiques.

### Funding

John R. Hutchinson was funded by BBSRC grants BB/C516844/1 and BB/H002782/1. Cyril Etienne and Alexandra Houssaye were funded by ERC grant ERC-2016-STG GRAVIBONE. Cyril Etienne was funded by a PhD fellowship from Centre de Recherches Interdisciplinaires and Université de Paris. The funders had no role in study design, data collection and analysis, decision to publish, or preparation of the manuscript.

### Grant Disclosures

The following grant information was disclosed by the authors:
BBSRC: BB/C516844/1 and BB/H002782/1.
ERC: ERC-2016-STG GRAVIBONE.
Centre de Recherches Interdisciplinaires and Université de Paris.

### Competing Interests

John R. Hutchinson is an Academic Editor for PeerJ.

### Author Contributions

- Cyril Etienne conceived and designed the experiments, performed the experiments, analyzed the data, prepared figures and/or tables, authored or reviewed drafts of the paper, and approved the final draft.
- Alexandra Houssaye conceived and designed the experiments, analyzed the data, authored or reviewed drafts of the paper, and approved the final draft.

- John R. Hutchinson conceived and designed the experiments, performed the experiments, analyzed the data, authored or reviewed drafts of the paper, and approved the final draft.

## Animal Ethics

The following information was supplied relating to ethical approvals (i.e., approving body and any reference numbers):

Ethical approval is not applicable. Only dead animals were used. Animals died of natural causes unrelated to the study.

## Data Availability

The raw measurements and the code used for the t-tests are available in the Supplemental Files.

## Supplemental Information

Supplemental information for this article can be found online at http://dx.doi.org/10.7717/peerj.11314#supplemental-information.

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
