# Peer review of "Limb myology and muscle architecture of the Indian rhinoceros Rhinoceros unicornis and the white rhinoceros Ceratotherium simum (Mammalia: Rhinocerotidae)"

_PeerJ, doi:10.7717/peerj.11314_

## Round 0.1 · original submission · Minor Revisions

Thank you very much for your very interesting article. I had been very much enjoyed reading this article.

We need some revisions to this article before acceptance. Please respond to all reviewer comments, point by point. I am looking forward to seeing your revision version.

·

Basic reporting

The authors have provided the first quantitative comparison of white and greater-one-horned rhinoceros appendicular musculature in both adult and neonate individuals, and have presented their results is a broad comparative context with regard to phylogenetically close relatives (horses and tapirs) and species of comparable mass (proboscidians, hippopotamids etc.). The article is written is good English throughout. There are a few grammatical errors, and some sections which would benefit from structural rearrangement, but on the whole this article is very well written and meets the expected standards of an academic article. There are a few minor changes needed to the references (some of which seem characteristic of mistakes in citation-plugin software), and rectifying these will not take long. The literature used does seem to cover the major comparative groups which inform the results and discussion; there may be room for the authors to expand upon this in certain sections (see .pdf comments). The article is well structured (with the exception of the first Discussion, which in my opinion belongs in the Results), with informative tables and figures. I have a few niggling points on the figures (see .pdf), though on the whole I think they are well laid out and highly informative. From what I can tell, all supporting information seems to be in order, and is beneficial and relevant to the submitted work. I would encourage the authors to look through their supplementary file (especially File S2.docx) to check the grammar – there are some things which I picked up on in the main document which also apply to the supplement, but nothing major. Some minor corrections to references are required, and possibly the addition of some supporting literature here and there.

Experimental design

The subject of this article falls comfortably within the remit of the journal PeerJ. The research question is well defined, with clear hypotheses. There is scope to increase support for the rationale behind the hypotheses (see .pdf), but in general they are sound, sensible and well explained. The experimental design is also well laid out, and meets the criteria expected for a comparative myology study. The methodologies are clearly explained with rationale for most of the calculations. Minor additional explanations would benefit the reader, which I have detailed in the .pdf. Overall this is a very sound example of methodology for a comparative myology study of large mammals.

Validity of the findings

There could be a case for this study being a replication of the work of Haughton (1867), but personally I do not buy into that. The results that the authors present here are novel, offering comparisons which have not been performed before across species and ontogeny in rhinoceroses, and the quantitative nature of the information produced (e.g. PCSA, Fmax) set it apart from previous dissection reports on rhinoceroses. Data provided are sound and methodology well explained, and to the best of my understanding the statistical methodologies are robust and suitable. The conclusions link well to the comparisons in the Discussion; there may be room for a little more expansion of the findings, perhaps in perspective of the comparative ecology of the rhinoceros species in question, or to the current understanding of “graviportality”. That said, I believe the conclusions are sound and a good summation of the work performed by the authors.

Additional comments

I very much enjoyed this article – as the authors point out themselves, this is a group which is well known by the general public but has been afforded somewhat less attention by the scientific community, especially when it comes to comparative functional anatomy. I therefore applaud the work done by the authors, who have clearly put a lot of effort into generating their results. My comments on the reporting are numerous, but minor (see attached .pdf). On the whole I think this is a very well written article, which will benefit from some changes to the terminology or the sentence structure here and there. I think my only major suggestion would be the repositioning of the first Discussion section into the Results, blending it with the original Results text as all the information which is presented is useful and interesting. There was a notable lack of citing tables and figures in useful places throughout the Results and Discussion sections, which again is a minor consideration which can be easily rectified. The experimental design is not revolutionary, but in this case it does not need to be, and the methods are well within the usual procedures for such studies – and these have been done to a very good standard in the article. The results and discussion sections are valid and bring in a good pool of comparative literature; there is certainly space to explore a little more in certain aspects, or add some information in the Introduction which would better inform the reader going into the Results and Discussion. My specific comments can be found on the attached .pdf document. I wish the authors well with this article, and hope to see more like it exposing comparative anatomy of popular but maligned taxonomic groups in the future.

Reviewer 2 ·

Basic reporting

The paper is well written, and the language is clear. References were adequately cited. The article is well-structured and figures and tables were of high quality.

Experimental design

The aim of the study is within the scope of the journal. Research question was well-defined and it is worth investigating. Methods were described with sufficient details.

Validity of the findings

The study provided data from one adult and one neonate for each species. To make the findings of the paper more robust, it is ideal that the authors can increase the number of the specimens studied, as inter-individual variabilities of the muscle parameters are known to be quite large. However, I understand that rhino specimens are very rare and it is extremely difficult to increase the number. So the small number of specimens studied is I think acceptable.

Additional comments

The manuscript provided, for the first time, qualitative descriptions and quantitative date of muscle architecture in two species of rhinos. The data are very unique and certainly contribute to our understanding of the functional morphology of terrestrial animals with large body mass. The only weakness is that the study provided data from only one adult for each species, even though inter-individual variabilities of the muscle parameters are known to be quite large. I know it is difficult to add new data, but I would like to see some more discussions about this limitation.

Reviewer 3 ·

Basic reporting

Overall, the manuscript is well-supported with the authors’ new data as well as background information and provides insights into the locomotion of large terrestrial vertebrates.

Experimental design

Overall, the experimental design is sound. Of the four hypotheses stated at the end of the introduction, the first two should be a bit more specific.

Validity of the findings

The anatomical description is detailed and quantitative data have been provided. The conclusions are supported by the data and relate back to the original stated hypotheses.

Additional comments

I thank the authors for the opportunity to review this interesting manuscript! I have some general comments and some line-by-line comments/questions.

In general, the first section of the discussion is a bit tedious. Since the general organization of this section parallels the ‘organization’ section of the Results, I suggest combining these into a section on anatomical description (in the results or as its own section). Statements about particular myological features/functional implications/interesting comparative aspects can stay in the discussion. Alternatively, some of the comparative information (rhinos vs other ungulates) could be summarized in a table or figure.

Related to this, the hypotheses get kind of lost in the muscle descriptions, as do the interesting conclusions. The way the discussion is laid out right now (limb by limb, region by region for both comparative and quantitative myology) doesn’t synthesize/address the results within the broader context of rhino anatomy/locomotion. Because the main thrust of the paper relates to adaptations for supporting a large body mass, the discussion could be streamlined/clarified by focusing on aspects of the myology that relate back to the original hypotheses.

This would also strengthen the conclusion by providing context for it that could be restated and expanded upon. The final bits of the conclusion where you discuss the unsuitable of the ‘graviportal’ grouping is super interesting and I think could be elaborated on in the discussion.

L79 – ‘tradeoff’ slightly confusing here. Maybe ‘advantage’ instead, or ‘the latter can generate greater force for the same muscle volume, demonstrating a tradeoff between excursion and force production’

L109 – replace ‘would be’ with ‘is’

L110-112 – awkward sentences here, can combine

L122-135 – this paragraph seems superfluous, since these citations will come up in the rest of the manuscript. You could make this more concise by simply referring to the species with data available (or not available) for comparison

L139-144 – rephrase the first two hypotheses, to provide more detail on the adaptations you expect to see. Something like ‘We hypothesize that rhino musculature will share features linked to fast running with their close relatives..’ and ‘we expect to find evidence of adaptation to large body size, similar to those observed in other heavy-bodied taxa…’

L142 – delete ‘in’ in the sentence ‘unlike in their cousins’

L204 – delete ‘compared’

L428 – This first section of the discussion feels ponderous.

L457-458 – this sentence provides functional info for context

L814-820 – the meaning of these sentences is unclear to me. What does ‘less biomechanical benefits’ mean? The SVC is more powerful than the SVT in horses, but the RHB, which is less powerful than both, is more biomechanically important in horses? The paragraph as a whole is a bit confusing.

L844 - 846 – Since you didn’t measure any tendon or elastic properties, is this relevant? Also substitute ‘recover’ for ‘restitute’

L893-901 – the focus of this seems to be the equine limb rather than your data – reframe this slightly so that the horse data/discussion is in direct comparison to the rhino data

L967 – perhaps ‘negative scaling relationship’ instead of ‘negative allometry’ since the nature of the relationship is hard to discern from small sample size?

L1019-1021 – restate the traits here and make sure they are explicitly discussed in the discussion

---

## Round 0.2 · Minor Revisions

There are a few comments that you need to address in your revision before we can accept this article. I am looking forward to receive your manuscript.

·

Basic reporting

The revised article is written in very good English throughout. There are one or two grammatical anomalies which I have highlighted, but on the whole this article is very well written and meets the expected standards of an academic article. Minor changes from the original manuscript have been comprehensively undertaken. The literature used covers the major comparative groups which inform the results and discussion. The article is well structured, with informative tables and figures. From what I can tell, all supporting information seems to be in order, and is beneficial and relevant to the submitted work.

Experimental design

The subject of this article falls comfortably within the remit of the journal PeerJ. The research question is well defined, with clear hypotheses. The experimental design is also well laid out, and meets the criteria expected for a comparative myology study. The methodologies are clearly explained with rationale for the calculations. Overall this is a very sound example of methodology for a comparative myology study of large mammals.

Validity of the findings

The results that the authors present here remain novel, offering comparisons which have not been performed before across species and ontogeny in rhinoceroses, and the quantitative nature of the information produced (e.g. PCSA, Fmax) set it apart from previous dissection reports on rhinoceroses. Data provided are sound and methodology well explained, and to the best of my understanding the statistical methodologies are robust and suitable. The conclusions link well to the comparisons in the Discussion, and there is now more expansion on the findings in perspective of comparative ecology and our current understanding of “graviportality”.

Additional comments

The authors have provided the first quantitative comparison of white and greater-one-horned rhinoceros appendicular musculature in both adult and neonate individuals, and have presented their results is a broad comparative context with regard to phylogenetically close relatives (horses and tapirs) and species of comparable mass (proboscidians, hippopotamids etc.). I very much enjoyed this article – as the authors point out themselves, this is a group which is well known by the general public but has been afforded somewhat less attention by the scientific community, especially when it comes to comparative functional anatomy. I therefore applaud the work done by the authors, who have clearly put a lot of effort into generating their results.
Having viewed this article twice now, I have very little to add to my previous appraisal. With regards to the revision documents and responses to the reviews, I agree with the rationale for all their rebuttals. I must also emphasise the benefit for the reviewer of providing the tracked changes very clearly as was done by the authors – well done! Regarding the revised manuscript, I have some additional but very minor suggestions (see attached .pdf). I think this is a very well written article, which has benefitted from changes to the terminology and the sentence structure from the previous version. My only major suggestion was taken on, and the authors have now married the Results and Discussion sections together very well. The methods remain well within the expected procedures for such a study. The results and discussion sections remain valid and bring in a good pool of comparative literature; they also flow much better now they have been rearranged. My (very minor) specific comments can be found on the attached .pdf document. I congratulate the authors once again on this article, and hope to see more like it exposing comparative anatomy of popular but maligned taxonomic groups in the future.

Reviewer 2 ·

Basic reporting

The paper is well written, and the language is clear. References were adequately cited. The article is well-structured and figures and tables were of high quality.

Experimental design

The aim of the study is within the scope of the journal. Research question was well-defined and it is worth investigating. Methods were described with sufficient details.

Validity of the findings

The study provided data from one adult and one neonate for each species. To make the findings of the paper more robust, it is ideal that the authors can increase the number of the specimens studied, as inter-individual variabilities of the muscle parameters are known to be quite large. However, I understand that rhino specimens are very rare and it is extremely difficult to increase the number. So the small number of specimens studied is I think acceptable.

Additional comments

The paper should be accepted.

---

## Round 0.3 · accepted · Accept

Congratulations on your accepted manuscript.